# Mapping Review of the Correlations Between Periodontitis, Dental Caries, and Endocarditis

**DOI:** 10.3390/dj13050215

**Published:** 2025-05-16

**Authors:** Mario Dioguardi, Ciro Guerra, Pietro Laterza, Gaetano Illuzzi, Diego Sovereto, Enrica Laneve, Angelo Martella, Lorenzo Lo Muzio, Andrea Ballini

**Affiliations:** 1Department of Clinical and Experimental Medicine, University of Foggia, Via Rovelli 50, 71122 Foggia, Italy; ciro_guerra.556675@unifg.it (C.G.); pietro_laterza.572581@unifg.it (P.L.); gaetano.illuzzi@unifg.it (G.I.); diego_sovereto.546709@unifg.it (D.S.); enrica.laneve@unifg.it (E.L.); lorenzo.lomuzio@unifg.it (L.L.M.); andrea.ballini@unifg.it (A.B.); 2DataLab, Department of Engineering for Innovation, University of Salento, 73100 Lecce, Italy; angelo.martella@unisalento.it

**Keywords:** endocarditis, periodontitis, dental caries, bacteraemia, cardiovascular diseases, oral health, antibiotic prophylaxis, oral microbiota

## Abstract

**Background/Objectives:** The correlation between cardiovascular diseases, particularly infective endocarditis, and oral disorders such as periodontitis and dental caries has been widely discussed in the scientific literature. In this mapping review, we aim to examine the available evidence on the link between these conditions, focusing on the pathogenetic mechanisms that underlie the development of endocarditis in patients with oral diseases. **Methods:** A systematic search was conducted across three major databases—PubMed, Scopus, and ScienceDirect—as well as grey literature in Google Scholar. Relevant articles were selected based on inclusion and exclusion criteria, focusing primarily on systematic reviews. The data extracted included study characteristics, main outcomes, and risk-of-bias evaluations. **Results:** A total of 13 systematic reviews were included in this mapping review. The findings suggest there is a significant connection between periodontal disease, dental caries, and the incidence of infective endocarditis. The evidence highlights that oral bacteria, particularly *Streptococcus* species, can enter the bloodstream during daily activities and invasive dental procedures, contributing to the development of endocarditis in susceptible individuals. However, the role of antibiotic prophylaxis in preventing endocarditis following dental procedures remains controversial. **Conclusions:** This review reinforces the importance of oral health in preventing cardiovascular complications, especially infective endocarditis. Although antibiotic prophylaxis may reduce the risk in high-risk individuals, further research is needed to clarify its effectiveness. Enhanced awareness of and education on the shared risks between oral and cardiovascular health could improve prevention strategies and patient outcomes.

## 1. Introduction

Over the years, scientific studies have highlighted the correlation between oral diseases and systemic conditions, particularly cardiovascular diseases and periodontitis. In fact, epidemiological evidence suggests there is a consistent association between periodontitis and an increased risk of developing cardiovascular disease [1].

Cardiovascular diseases (CVDs) are a group of disorders that affect the heart and/or the body’s entire vascular system. Most cardiovascular diseases are chronic conditions that persist for extended periods. However, in some cases, they may manifest as acute episodes, such as myocardial infarction and stroke, which occur suddenly due to the obstruction of blood vessels supplying the brain or heart [2].

Cardiovascular diseases occur more frequently in individuals who smoke, have hypertension, have elevated cholesterol levels (particularly low-density lipoprotein cholesterol (LDL-C)), are overweight, do not engage in physical activity, and/or suffer from diabetes [3].

Among cardiovascular diseases, endocarditis most commonly has a bacterial aetiology and can present in conjunction with other infectious conditions such as periodontitis or dental caries, sharing certain pathogens or risk factors.

The term periodontal disease refers to a group of inflammatory conditions that affect both the superficial and deep supporting tissues of the tooth [4]. In the former case, gingivitis is present; however, when the inflammation extends beyond the gingival region and involves the underlying areas that constitute the supporting tissues of the tooth, periodontitis or chronic periodontitis occurs, progressively destroying the tooth-supporting structures. It typically manifests as a worsening of gingivitis and, if left untreated, leads to tooth mobility and eventual tooth loss. Other symptoms are rare, except when abscesses develop; in these cases, pain and swelling are common. Diagnosis is based on inspection, periodontal probing, and intraoral radiographs. Treatment involves scaling below the gum line and a home hygiene regimen [5].

Caries is a dental disease characterised, in its initial phase, by the destruction of the enamel caused by certain bacterial species, particularly *Streptococcus mutans* [6]. If left untreated, caries progresses to affect the deeper layers of the tooth, extending to the dentine and eventually reaching the pulp and roots. In addition to significant painful symptoms, untreated caries can lead to the loss of the affected tooth and may require antibiotics and surgery. Cariogenic disease is a complex and multifactorial condition that, despite the implementation of various preventive methods, remains one of the most common diseases among humans. The development of dental caries is due to the interaction between genetic and environmental factors, which influence each other [7].

Several mechanisms have been proposed to explain the correlation between oral infections and cardiovascular conditions such as endocarditis. Among these, chronic inflammation caused by periodontitis or caries may induce systemic inflammatory responses, promoting endothelial dysfunction and creating a favourable environment for microbial colonisation. Furthermore, repeated episodes of transient bacteraemia—triggered by daily oral activities or invasive dental procedures—enable oral bacteria, particularly *Streptococcus* species, to reach and adhere to damaged cardiac tissues. The virulence factors of these bacteria, combined with an impaired immune response or pre-existing cardiac lesions, may facilitate the formation of vegetations on heart valves, ultimately leading to infective endocarditis. These mechanisms underline the importance of exploring the pathogenic interplay between oral and cardiovascular health, as discussed in detail in the following sections [8].

The aim of this mapping review is to examine the available scientific literature on the correlation between cardiovascular diseases, particularly endocarditis, and oral disorders, with a focus on periodontitis and, secondarily, cariogenic disease [9].

This review will focus on the analysis of existing evidence regarding the correlations between periodontal disease, caries, and endocarditis. The objective of the review is to identify the common pathogenetic mechanisms between endocarditis and oral disorders, with particular emphasis on the effects of chronic inflammation, epigenetic modifications, and immune responses. The review will also examine the links between bacteraemia caused by oral infections and the worsening of cardiac conditions such as endocarditis. An additional aim of this review will be to map the areas of research explored and identify gaps in the existing literature. This will enable the suggestion of future directions for further investigation into the connections between oral health and cardiovascular diseases, with a particular focus on new potential therapeutic approaches.

## 2. Materials and Methods

### 2.1. Protocol and Registration

This mapping review was prepared according to the guidelines of the SANRA (Scale for the Assessment of Narrative Review Articles) [10]. We decided not to register the review protocol, despite it being prepared prior to conducting the database search.

### 2.2. Eligibility Criteria

All studies examining the correlations between cardiovascular diseases, particularly endocarditis, and oral disorders were considered potentially eligible. Special attention was given to systematic literature reviews. Given the vast scope of this topic, focusing on systematic reviews was a strategic decision aimed at reducing the volume of information, allowing us to concentrate on the aspects and results that had already been extensively evaluated in previous reviews, without wasting time and resources synthesising the results of each individual study linking oral medicine with cardiovascular diseases (forming part of a considerable number of studies). No restrictions were applied based on the year of publication or language, provided that an abstract in English was available.

The exclusion criteria applied included: (i) narrative reviews, editorials, letters to the editor, and case reports; (ii) studies that did not address the association between oral diseases and endocarditis or cardiovascular diseases; (iii) duplicate publications; and (iv) reviews lacking sufficient methodological transparency or a clear description of the included studies. Only systematic reviews were considered for inclusion in order to ensure a high-quality synthesis of evidence.

### 2.3. Information Sources

For the preparation of this review, three databases were used: PubMed, Scopus, and ScienceDirect. Additionally, grey literature searches were conducted on Google Scholar. Potentially eligible articles were also sought within the reference lists of literature reviews on the correlations between cardiovascular diseases and oral disorders. The search was conducted between the 1st February 2025 and 10th February 2025, with the last update of the identified records made on 14th February 2025.

### 2.4. Search

The research questions that guided the selection of and search for studies were as follows:

What is the scientific evidence on the correlations between endocarditis and oral disorders, particularly periodontitis and dental caries?

How do common inflammatory, genetic, and epigenetic mechanisms contribute to the relationship between these conditions and oral health?

What gaps exist in the literature regarding the interaction between oral health and endocarditis, and what future directions could be explored to further our understanding of this connection?

The authors responsible for the study search used the following keywords in PubMed: Search: (oral OR dental OR parodontal OR caries OR periodontitis) AND (endocarditis) Sort by: Most Recent (“mouth”[MeSH Terms] OR “mouth”[All Fields] OR “oral”[All Fields] OR (“dental health services”[MeSH Terms] OR (“dental”[All Fields] AND “health”[All Fields] AND “services”[All Fields]) OR “dental health services”[All Fields] OR “dental”[All Fields] OR “dentally”[All Fields] OR “dentals”[All Fields]) OR (“parodont”[All Fields] OR “parodontal”[All Fields] OR “parodontitis”[All Fields]) OR (“carie”[All Fields] OR “dental caries”[MeSH Terms] OR (“dental”[All Fields] AND “caries”[All Fields]) OR “dental caries”[All Fields] OR “caries”[All Fields]) OR (“periodontal”[All Fields] OR “periodontally”[All Fields] OR “periodontically”[All Fields] OR “periodontics”[MeSH Terms] OR “periodontics”[All Fields] OR “periodontic”[All Fields] OR “periodontitis”[MeSH Terms] OR “periodontitis”[All Fields] OR “periodontitides”[All Fields])) AND (“endocarditis”[MeSH Terms] OR “endocarditis”[All Fields] OR “endocarditides”[All Fields]).

Translations

oral: “mouth”[MeSH Terms] OR “mouth”[All Fields] OR “oral”[All Fields]

dental: “dental health services”[MeSH Terms] OR (“dental”[All Fields] AND “health”[All Fields] AND “services”[All Fields]) OR “dental health services”[All Fields] OR “dental”[All Fields] OR “dentally”[All Fields] OR “dentals”[All Fields]

parodontal: “parodont”[All Fields] OR “parodontal”[All Fields] OR “parodontitis”[All Fields]

caries: “carie”[All Fields] OR “dental caries”[MeSH Terms] OR (“dental”[All Fields] AND “caries”[All Fields]) OR “dental caries”[All Fields] OR “caries”[All Fields]

periodontitis: “periodontal”[All Fields] OR “periodontally”[All Fields] OR “periodontically”[All Fields] OR “periodontics”[MeSH Terms] OR “periodontics”[All Fields] OR “periodontic”[All Fields] OR “periodontitis”[MeSH Terms] OR “periodontitis”[All Fields] OR “periodontitides”[All Fields]

endocarditis: “endocarditis”[MeSH Terms] OR “endocarditis”[All Fields] OR “endocarditides”[All Fields].

During the search process, filters were applied when available in order to refine the results and include review articles. In PubMed, the filter “Systematic review” was selected to prioritise high-quality evidence. No language restrictions were applied, as long as an abstract was available in English. The search was limited to human studies.

A total of 3963 records were identified in PubMed. In Scopus and ScienceDirect, the following keywords were used: TITLE-ABS-KEY ((oral OR dental OR periodontal OR caries OR periodontitis) AND (endocarditis)). Consequently, 5587 records were identified through Scopus, and 36,821 records were found using ScienceDirect.

The search for eligible articles and reports was a collaborative effort between two reviewers, M.D. and D.S. After reaching a consensus on the eligibility criteria, keywords, and selected databases, the primary reviewers independently conducted the search for articles and reports [11]. They meticulously recorded the number of articles retrieved for each keyword and from each designated database. Duplicate studies identified across different databases were systematically removed using EndNote 8 software (Philadelphia, PA, USA).

The authors carefully carried out the removal of study overlaps that could not be processed using EndNote during the screening phase [12]. Subsequently, the two reviewers proceeded with screening and inclusion of studies, engaging in comparative analyses and constructive discussions to determine which studies to include in the mapping review [13].

### 2.5. Synthesis of Results

The studies deemed eligible were read, and the key information was noted and used to draft this mapping review.

The data to be extracted from the included articles were predetermined by the two reviewers and included the first author of the study, the publication date, the country where the research was conducted, the number of studies included, the registration protocol number, the main outcome with the principal result, the presence of meta-analysis, and any evaluation of the risk of bias. The data were extracted and recorded in a dedicated table.

### 2.6. Risk of Bias

The risk of bias in the individual systematic reviews was assessed by two authors (M.D. and A.B.). The ROBIS (Risk of Bias in Systematic Reviews) was used as an assessment tool specifically developed to assess the risk of bias in systematic reviews. Studies with a high risk of bias were excluded from the review [14].

The distinction between bias within the review process (meta-bias) and bias in the primary studies included in the review is critical. A systematic review may be considered to have a low risk of bias even if all the included primary studies have a high risk of bias, provided that the review appropriately assesses the risk of bias in the primary studies before drawing its conclusions [14].

## 3. Results

The search conducted using the databases yielded a total of 9550 articles. After duplicates were removed using EndNote X8 software, the number of articles reduced to 6889. Applying inclusion and exclusion criteria based on titles, abstracts, or keywords, we identified approximately 54 articles. Given the abundance of studies, it was deemed appropriate to exclude articles considered to be of lesser qualitative significance and to focus solely on systematic reviews.

A full-text assessment was subsequently conducted. During this phase, 44 studies were excluded for the following main reasons:✓A total of 27 did not specifically address the correlation between endocarditis and oral diseases but rather discussed broader cardiovascular implications or unrelated infections (A);✓Five lacked sufficient methodological transparency, either due to the absence of protocol registration or the presence of poorly defined protocols and unclear inclusion criteria (B);✓Six were duplicate systematic reviews or had substantial overlaps with previously published reviews, which the reviews in question effectively updated (C);✓Five were not systematic reviews but clinical guidelines, position papers, or survey-based studies (D) (Table 1).

As a result, 11 systematic reviews were included in the final analysis.

The entire selection process is represented in the flow chart (Figure 1).

The results were extracted and presented in Table 2.

In the 11 systematic reviews included, a total of approximately 157 primary studies were analysed. However, due to overlaps between reviews, many of these studies were included in more than one review. For example, common references were identified in the reviews by Sperotto et al. (2024) [68], Cahill et al. (2017) [161] and Lafaurie et al. (2019) [127], González Navarro et al. (2017) [50], and others, particularly regarding the incidence of bacteraemia and the effects of antibiotic prophylaxis. This overlap introduces a potential source of bias, as some studies may be over-represented in the evidence synthesis. Fortunately, the included reviews provided detailed lists of the primary studies analysed, limiting the possibility of the precise quantification of duplications, as the total number of primary studies included in the review was 101 (Lacassin et al., 1995 [59]; Strom et al., 1998 [60]; Porat et al., 2008 [61]; Chen et al., 2015 [62]; Chen et al., 2018 [63]; Tubiana et al., 2017 [64]; Thornhill et al., 2023 [65]; Thornhil et al., 2022 [66]; Thornhill et al., 2024 [67]; Keller et al., 2017 [69];van den Brink et al., 2017 [70]; Bates et al., 2017 [71]; Sakai Bizmark et al., 2017 [72]; Garg et al., 2019 [73]; Quan et al., 2020 [74]; Vähäsarja et al., 2020 [75]; Bikdeli et al., 2013 [76]; DeSimone et al., 2015 [77]; Toyoda et al., 2017 [78]; Sun et al., 2017 [79]; Thornhill et al., 2022 [80]; Rogers et al., 2008 [81]; Pasquali et al., 2015 [82]; Pant et al., 2015 [83]; Thornhill et al., 2017 [84]; DeSimone et al., 2021 [85]; Mackie et al., 2016 [86]; Knirsch et al., 2020 [87]; Weber et al., 2022 [88]; Krul et al., 2015 [89]; Duval et al., 2012 [90]; Dayer et al., 2015 [91]; Shah et al., 2020 [92]; Zegri-Reiriz et al., 2018 [94]; Thornhill et al., 2018 [95]; Khairat, 1966 [97]; Shanson et al., 1985 [98]; Maskell et al., 1986 [99]; Shanson et al., 1987 [100]; Vergis et al., 2001 [101]; Lockhart et al., 2004 [102]; Diz Dios et al., 2006 [103]; Abu-Ta’a et al., 2008 [104]; Anitua et al., 2008 [105]; Lockhart et al., 2008 [106]; Asi et al., 2010 [107]; Esposito et al., 2010 [108]; Siddiqi et al., 2010 [109]; Chandramohan et al., 2011 [110]; Maharaj et al., 2012 [111]; DuVall et al., 2013 [112]; Limeres Posse et al., 2016 [113]; Van der Meer et al., 1992 [116]; Pourmoghaddas et al., 2018 [118]; Ali et al., 2017 [119]; Cantekin et al., 2015 [120]; Cantekin et al., 2013 [121]; Suma et al., 2011[122]; Siahi-Benlarbi et al., 2010 [123]; da Fonseca et al., 2009 [124]; Tasioula et al., 2008 [125]; Stecksén-Blicks et al., 2004 [126]; Snanson et al., 1978 [128]; Roberts et al., 1987 [129]; Hall et al., 1993 [130]; Hall et al., 1996 [131]; Wahlmann et al., 1999 [132]; de Souza et al., 2016 [134]; Deppe et al., 2007 [135]; Wu et al., 2008 [136]; Hakeberg et al., 1999 [137]; Nakamura et al., 2011 [138]; Bratel et al., 2011 [139]; Lockhart et al., 2009 [141]; Bahrani-Mougeot et al., 2008 [142]; Heimdahl et al., 1980 [143]; Rajasuo et al., 2004 [144]; Roberts et al., 2006 [145]; Tomás et al., 2007 [146]; Roberts et al., 1998 [147]; Benítez-Páez et al., 2013 [148]; Maharaj et al., 2012 [149]; Peterson et al., 1976 [150]; Rahn et al., 1995 [151]; Sefton et al., 1990 [152]; Hall et al., 1996 [153]; Tuna et al., 2012 [154]; Sweet et al., 1978 [155]; Cannell et al., 1991 [156];Aitken et al., 1995 [157]; Josefsson et al., 1985 [158]; Tomás et al., 2008 [159]; Piñeiro et al., 2010 [160]; Imperiale and Horwitz, 1990 [162]; Baltch et al., 1982 [163]; Coulter, et al., 1990 [164]; Head et al., 1984 [165];; Horstkotte et al., 1987 [166]; Thornhill et al., 2011 [167]; Desimone et al., 2012 [168]; and Salam et al., 2014 [169]).

The primary outcomes evaluated in the included systematic reviews were as follows:✓Incidence of IE following dental procedures;✓Association between oral conditions (periodontitis and dental caries) and IE;✓Bacteraemia induced by dental procedures;✓Effectiveness of AP in preventing IE;✓Antimicrobial resistance patterns;✓Impact of oral health interventions in high-risk cardiac patients.

These outcomes are summarised in Table 3 and further discussed in the Discussion section. In total, 8 out of 11 reviews included outcomes related to antibiotic prophylaxis, while 4 out of 11 specifically quantified its effectiveness. Only 1 out of 11 reviews included data on caries-related infections, focusing specifically on children with congenital heart disease (CHD).

Among the reviews that reported data on AP, the most commonly studied regimens included oral or intravenous amoxicillin with clavulanic acid, and oral azithromycin for penicillin-allergic patients. According to Lafaurie et al. (2019) [127], azithromycin showed greater efficacy in reducing bacteraemia compared to clindamycin. Amoxicillin was typically administered as a single 2 g oral dose 30–60 minutes prior to the procedure, whereas intravenous regimens were reserved for hospitalised or high-risk patients undergoing surgery.

Barbosa-Ribeiro et al. (2024) observed high levels of resistance to clindamycin, metronidazole, and rifampicin in E. faecalis, highlighting the importance of local antibiogram data [24]. Only four of the included reviews provided direct comparisons between antibiotic regimens, and no clear consensus emerged from the literature regarding the optimal agent or dosage.

Preventive antibiotics were discussed primarily in relation to invasive procedures [170] and not as a routine treatment for caries or mild periodontal therapy.

Regarding caries, only one review (Karikoski et al., 2021) addressed this condition, showing that there is a significantly higher prevalence of dental caries among children with congenital heart disease (CHD) [117]. However, this review did not evaluate the role of antibiotics in caries management as a preventive strategy for IE. No systematic review supported the routine use of antibiotic therapy for non-invasive caries treatment (e.g., restorations).

As for periodontal treatment, several reviews (Kussainova et al., 2025 [58]; Lafaurie et al., 2019 [127]; González Navarro et al., 2017 [140]) considered procedures such as scaling and periodontal surgery in the context of bacteraemia risk. However, the findings consistently showed no significant association between non-invasive periodontal treatment and an increased risk of IE (e.g., Kussainova et al. reported an OR of 0.69; *p* = 0.41 for periodontal therapy) [58]. Consequently, routine antibiotic prophylaxis for periodontal therapy is not supported by current evidence, unless the procedure is invasive and the patient is considered high-risk.

Age-related findings were limited. Only one review (Karikoski et al., 2021) specifically focused on children with congenital heart disease and reported a higher caries burden and associated IE risk [117]. Other reviews included populations with heterogeneous age ranges but did not stratify findings by age group, making it difficult to assess whether age modifies the association between oral diseases and IE.

### Risk of Bias

The risk of bias for systematic reviews was determined using the ROBIS tool, and for each factor, it was evaluated as “low”, “high”, or “unclear”. The three phases of the evaluation process were as follows: Phase 1—the evaluation of the relevance of the research question (PICO); Phase 2—the identification of the critical points of the review process; and Phase 3—the evaluation of the overall risk of bias of the review. All data related to the risk of bias are reported in Table 4.

The main critical issues related to the individual revisions are as follows:✓Sperotto et al., 2024 [68]: Regarding the identification and selection of studies (?), the start or end dates of the review were not specified.✓Friedlander and Couto-Souza, 2023 [93]: Regarding the study eligibility criteria (?), the protocol number with which the systematic review was registered was not reported. Regarding data collection and study appraisal (?), the risk of bias was not formally assessed using an appropriate scale or tool. Regarding the dentification and selection of studies (?), the selection was performed using only one database (PubMed).✓Albakri et al., 2022 [96]: Regarding data collection and study appraisal (?), the risk of bias was not formally assessed using an appropriate scale or tool.✓Karikoski et al., 2021 [117]: Regarding the dentification and selection of studies (?), the systematic review by Karikoski et al. (2021) [117] did not report registration of a review protocol in any public registry.✓González Navarro et al., 2017 [140]: Regarding the dentification and selection of studies (?), the systematic review by González Navarro et al. did not report registration of a review protocol in any public registry. Regarding data collection and study appraisal (?), the risk of bias was not formally assessed using an appropriate scale or tool.✓Cahill et al., 2017 [161]: Regarding the identification and selection of studies (?), the systematic review by Cahill et al. did not report registration of a review protocol in any public registry. Regarding data collection and study appraisal (?), the risk of bias was not formally assessed using an appropriate scale or tool.

## 4. Discussion

### 4.1. Association Between Endocarditis and Periodontitis

Endocarditis is an inflammation of the thin inner lining of the heart (endocardium) and the heart valves. Endocarditis most commonly affects the heart valves but can also form in shunts or other abnormal connections between the heart chambers [171].

In most cases, endocarditis is caused by an infection, although in a smaller percentage of cases, it has a non-infectious aetiopathogenesis. Infective endocarditis is generally of bacterial origin, but other pathogens, such as fungi, can also trigger the inflammatory process [172].

Bacterial endocarditis occurs when microorganisms from other parts of the body, such as the skin, oral cavity, intestines, or urinary tract, spread through the bloodstream and reach the heart [173].

Under normal conditions, the immune system recognises and defends the body against infectious agents, which—even if they reach the heart—would typically be harmless, passing through without causing an infection. However, if the cardiac structures are damaged, due to rheumatic fever, congenital defects, or other diseases, they may be susceptible to microbial invasion. In these conditions, it is easier for bacteria that have entered the body through the bloodstream to settle in the inner lining of the heart, overcoming the normal immune response to infections [174]. When the conditions are ideal, the infectious agents can organise themselves into aggregations known as “vegetations” (characteristic lesions of bacterial endocarditis) at the site of infection, whether it be a heart valve or other cardiac structures, including implanted devices [175]. There is a risk that these cell masses behave similarly toward blood clots, blocking the blood supply to organs, leading to heart failure or triggering a stroke [176].

Under microscopic examination, these vegetations reveal the presence of microcolonies of infecting microorganisms, embedded in a network of platelets, fibrin, and a few inflammatory cells [176].

The interaction between predisposing factors in the host and the inability of the immune system to eradicate the infectious agent from the endocardium makes the patient susceptible to infection. Bacterial endocarditis occurs when infectious agents enter the bloodstream and manage to “stick” to the heart tissue, multiplying in damaged or surgically implanted heart valves. This damaged tissue in the endocardium provides the ideal environment for the infecting agents to settle: the cardiac surface offers them the support they need to adhere and proliferate [177].

Not all bacteria that enter the bloodstream can cause endocarditis. Only infectious agents that have a tropism for valvular structures and endocardial tissues, meaning those that can interact with the surface of the heart lining and abnormal valves can potentially lead to the clinical picture of endocarditis [178].

If endocarditis is neglected, inflammation can damage or destroy endocardial tissue or heart valves, leading to life-threatening complications. If a heart defect is present, certain medical procedures can lead to transient bacteraemia, potentially responsible for endocarditis; these include tonsillectomy, adenoidectomy, intestinal and respiratory surgery, cystoscopy, bronchoscopy, colonoscopy, etc. There is also a risk of endocarditis when a patient undergoes certain dental procedures. In fact, the earliest studies on bacteraemia associated with the oral activity focused on the transient effects of daily oral activity as well as clinical interventions [80].

Microorganisms can spread through the blood and reach the heart via various pathways, some of which are related to dental and oral health activities, including daily actions such as brushing teeth, chewing, or other activities involving the mouth that can allow bacteria to enter the bloodstream [179], especially in the presence of periodontitis or gingivitis. In fact, microorganisms can spread from pre-existing infections, such as endodontic abscesses (a clear complication of penetrating caries in the pulp chamber) [180] or during periodontal disease, circulating in the bloodstream and settling in the endocardium if they encounter a favourable environment [181].

Alternatively, they can spread through dental medical procedures, such as tooth extractions, root planing, and scaling, which can introduce bacteria into the bloodstream, particularly if they cause bleeding [182].

### 4.2. Correlation with Periodontitis

Numerous studies have assessed the correlation between infective endocarditis and periodontitis, demonstrating that the association between these two conditions is primarily due to the spread of bacteria present in the oral cavity through the bloodstream [183].

It is well known that infective endocarditis is a very aggressive disease with high morbidity and mortality, and its connection with oral bacteria has raised continuous concerns among dentists, patients, and cardiologists [184]. The microbiota of the mouth is extremely diverse, with specific sites in the mouth such as the tongue, palate, cheek, teeth, and periodontal pockets each harbouring their own microbiota [185]. However, the main microorganisms responsible for the formation of dental plaque, which can lead to periodontitis, are oral streptococci belonging to the *viridans* group (such as *Streptococcus mutans* and *Streptococcus sanguis*) [186]. Since these streptococci form part of the dental plaque, they can enter the bloodstream, causing bacteraemia through daily activities such as chewing or brushing teeth [187].

A recently proposed causal model suggests that early bacteraemia can influence the endothelial surface of the heart for many years, promoting thickening of the valve and making it susceptible to vegetation from a subsequent infection, which could rapidly evolve into a fulminant infection [188]. In most studies, significantly higher rates of bacteraemia in patients with periodontitis compared to healthy patients have indicated that periodontitis serves as a gateway for oral streptococci to enter the bloodstream [189].

Periodontal infections are inflammatory diseases. Inflammation of the periodontal tissue leads to the deepening of the gingival sulcus and the formation of periodontal pockets, acting as a reservoir for a large number of microorganisms. Periodontal infections affect the oral tissues, and bacteria can enter the bloodstream through the inflamed, ulcerated sulcus and the epithelium of the pocket, as well as the adjacent gingival microcirculation [190].

Invasive dental procedures and routine daily activities such as chewing and tooth brushing are significant predisposing factors with respect to microorganisms entering the bloodstream [191]. Additionally, recent studies have shown that smoking increases the frequency and severity of periodontal disease. Smoking is associated with poor oral hygiene, and thus smokers tend to have greater amounts of plaque and tartar accumulation, along with increased susceptibility to bacterial growth [192]. Smoking also impacts periodontal disease through various systemic effects, such as reduced chemotaxis and phagocytosis by both oral and peripheral neutrophils as well as decreased antibody production, which favours the breakdown of periodontal tissues. Consequently, smokers with chronic periodontitis experience greater attachment loss and bone loss, more involvement of the root furcations, and deeper pockets, leading to an increase in bacteraemia induced by periodontitis [193].

It has been observed that effective treatment of periodontal infections is important in reducing local inflammation and bacteraemia. Furthermore, poor periodontal health seems to increase the risk of cardiovascular diseases. Indeed, good oral hygiene and prophylaxis for high-risk patients undergoing dental procedures are included among the various recommendations of the American Heart Association and the European Society of Cardiology [194].

In light of these findings, there has been strong interest in the role of the oral microbiota in cardiovascular diseases, making this topic a subject of ongoing study, especially regarding how antimicrobial treatments can generate drug-resistant mutant bacteria, which has become a significant social problem [195].

### 4.3. Correlation with Dental Caries

Although periodontitis has been studied more extensively in relation to infective endocarditis [118], the potential role of dental caries and its complications must be taken into consideration. Dental caries, especially if untreated, can progress to pulpitis and subsequently apical periodontitis, both of which are associated with local inflammatory responses with possible systemic implications. These infections can lead to abscess formation and contribute to transient or prolonged episodes of bacteraemia, particularly with the involvement of pathogens such as *Streptococcus mutans* [196], *Lactobacillus* spp. [197], and *Enterococcus faecalis* [198], which have been implicated in cases of infective endocarditis.

Some epidemiological studies have identified a higher prevalence of dental caries and untreated pulp infections in patients with congenital heart disease and a history of endocarditis [199]. The caries formation process itself may not directly cause endocarditis, but it creates a portal of entry for bacteria into the bloodstream, especially when accompanied by inadequate oral hygiene or during procedures such as endodontic therapy [200] and tooth extractions [201]. Furthermore, caries-related pulp necrosis often requires surgical intervention, increasing the risk of bacteraemia in susceptible individuals.

Therefore, although the evidence linking dental caries to endocarditis is less robust than that for periodontitis, the literature suggests that untreated caries and secondary infections should be considered significant risk factors, particularly for high-risk cardiac patients. This highlights the importance of early diagnosis, prevention, and management of caries as part of cardiovascular risk reduction strategies.

### 4.4. Evidence from the Literature

The results of a systematic review conducted by Kussainova et al. in 2025 identified an increased incidence of infective endocarditis following invasive dental procedures, such as tooth extractions and oral surgery [202], with the post-extraction socket and surgical wounds being more susceptible to bacteraemia caused by bacteria such as Streptococci, which are also responsible for endocarditis [58]. The main causative pathogens involved are *Staphylococcus aureus* (33.4%) and *Streptococcus* species, including *S. bovis* and *viridans* (32.0%), followed by *enterococci*, according to the data reported by Budea et al. (2022) [28].

The peak of bacteraemia occurs approximately five minutes after the dental procedure and decreases over time. The procedures associated with the highest incidence of bacteraemia, according to Martins et al. (2024) [21], are tooth extractions (62–66%), followed by scaling and root planing (36–44%), and oral health procedures such as dental prophylaxis and dental probing (27–28%) [203].

However, in a 2017 publication, González Navarro et al. [140] noted that antimicrobial prophylaxis prior to invasive dental procedures does not prevent bacteraemia but may reduce its extent and persistence [140].

Furthermore, in a 2022 publication, Albakri et al. [96] suggested that post-procedure bacteraemia may not be a reliable marker for infective endocarditis, pointing out that there are few studies investigating a direct association between antibiotic prophylaxis and infective endocarditis [96].

It is generally thought that most cases of bacterial endocarditis caused by oral bacteria are likely the result of frequent bacteraemia arising from daily activities such as tooth brushing, especially in the case of periodontitis, as supported by a 2017 study by Cahill et al. [161], although some cases may still stem from infrequent invasive dental procedures.

Moreover, the effectiveness of antibiotic prophylaxis for the prevention of infective endocarditis following invasive dental procedures remains unclear.

In fact, another systematic review of the literature conducted by Sperotto et al. (2024) highlighted that the use of antibiotic prophylaxis is associated with a reduced risk of infective endocarditis in high-risk individuals following invasive dental procedures, while no association was found for those at low/unknown risk [68].

Concerning endodontic procedures [204], especially in the presence of persistent endodontic infections, the most effective drugs according to the meta-analysis conducted by Barbosa-Ribeiro et al. (2024) [24] remain amoxicillin and clavulanic acid, followed by amoxicillin and benzylpenicillin, while there remain significant resistances against *E. faecalis* to clindamycin, gentamicin, metronidazole, and rifampicin [24]. The use of antibiotic prophylaxis is also strongly suggested in cases of bicuspid aortic valve or mitral valve prolapse [93].

While intravenous amoxicillin plus clavulanic acid may be administered to high-risk patients undergoing more invasive dental procedures requiring general anaesthesia, for penicillin-allergic patients, oral azithromycin has shown greater efficacy in reducing bacteraemia compared to clindamycin, according to the data reported by Lafaurie et al. in 2019 [127].

Bergadà-Pijuan et al. (2023) [114], however, argue that the evidence supporting or discouraging the use of antibiotic prophylaxis before dental procedures as a prevention measure for infective endocarditis is very limited [115].

Furthermore, there is no definitive scientific evidence, as stated by Lockhart et al. in 2019 [133], suggesting that performing dental procedures before cardiac surgery, such as cardiac valve surgery or left-ventricular assist device implantation, reduces the risk of mortality or comorbidities such as endocarditis.

In a 2021 publication, Karikoski et al. [117] explored the potential implications for children with congenital heart disease who had developed endocarditis and carious lesions, investigating the prevalence of carious lesions and concluding that there is a correlation between carious lesions and congenital heart failure. Additionally, the risk of infective endocarditis is increased among children with congenital heart failure, for whom endocarditis prophylaxis is recommended.

Several factors can predispose children with congenital heart failure to developing caries during their early years. Congenital disorders may affect enamel, predisposing the children to caries due to hypomineralisation defects [205]. Increased food intake to meet energy needs leads to the consumption of sugars and sugary liquids, often supplemented with additional meals at night, as well as a greater predisposition to infections. Diuretic use also leads to reduced salivation and a buffering effect on the acidity produced by bacteria [117].

### 4.5. Endocarditis Prophylaxis for At-Risk Patients Undergoing Non-Surgical Periodontal Treatment: Role of Amoxicillin

The use of AP to prevent IE in patients undergoing dental procedures remains a subject of ongoing debate, particularly in the context of non-surgical periodontal treatments such as scaling and root planing. Although a direct causal relationship between procedure-induced transient bacteraemia and the onset of IE has not been definitively established, a substantial body of evidence supports the notion that manipulations involving gingival or periapical tissues may introduce oral bacteria into the bloodstream, posing a potential risk for individuals with specific predisposing cardiac conditions (Albakri et al., 2022 [96]; Rutherford et al., 2022 [115]; and Lafaurie et al., 2019 [127]).

#### 4.5.1. Risk Associated with Non-Surgical Periodontal Procedures

Several studies have demonstrated that non-surgical periodontal procedures—particularly in regard to patients with active gingival inflammation or periodontitis—can induce bacteraemia at frequencies comparable to those observed following dental extractions. For instance, the study by Albakri et al. from 2022 [96] reported that periodontal probing and scaling were associated with bacteraemia rates ranging from 10% among patients with gingivitis to 40% among those with periodontitis, while tartar removal (i.e., supragingival and subgingival scaling) resulted in a bacteraemia prevalence of 24.5%. According to Albakri et al. [96], these variations are influenced by the severity of the underlying periodontal disease.

Conversely, the meta-analysis by Kussainova et al. from 2025 [58] found no significant association between invasive dental procedures and the development of infective endocarditis. Specifically, the pooled odds ratios (OR) were as follows: scaling—OR, 1.00; 95% CI, 0.85–1.18; *p* = 1.00; I^2^ = 0%; endodontic treatment—OR, 1.04; 95% CI, 0.73–1.49; *p* = 0.82; I^2^ = 0%; periodontal treatment—OR, 0.69; 95% CI, 0.28–1.67; *p* = 0.41; I^2^ = 69%.

Among the studies included in Kussainova et al.’s review [58], only that conducted by Lacassin et al. (1995) [59] reported a trend toward an increased risk associated with scaling and root canal therapy.

These findings indicate that even non-surgical procedures such as scaling and periodontal probing can induce transient bacteraemia, which represents a theoretical risk for infective endocarditis in predisposed patients. Consequently, such interventions may reasonably be classified as invasive procedures in the context of IE prevention for high-risk individuals.

#### 4.5.2. International Guidelines and Indications for Prophylaxis (Amoxicillin)

Current guidelines from the American Heart Association (AHA) [206] and the European Society of Cardiology (ESC) [194] recommend AP exclusively for patients at high risk—such as individuals with prosthetic heart valves, a history of IE, or complex congenital heart disease—who undergo dental procedures involving the manipulation of gingival tissues or the periapical region of teeth. Non-surgical periodontal treatments, particularly in the presence of active inflammation or gingival bleeding, clearly fall within this definition.

Most international protocols recommend a single dose of 2 g of oral amoxicillin without clavulanic acid, administered 30–60 minutes prior to the procedure, as reported in trials included in the reviews by Rutherford et al. (2022) [115], Lafaurie et al. (2019) [127] and Bergadà-Pijuan et al. (2023) [114].

While it is well established that antibiotic prophylaxis reduces post-procedural bacteraemia—as shown in the systematic review by Albakri et al. from 2022 [96], which reported an approximate 49% reduction in bacteraemia risk (Risk Ratio: 0.51, 95% CI: 0.45–0.58; *p* < 0.0001)—other studies suggest that this reduction does not necessarily translate into a decreased risk of developing IE. Supporting this view is the Cochrane systematic review by Rutterford et al. (2022) [115], which included non-surgical periodontal procedures (such as supragingival and subgingival scaling as well as curettage) among the evaluated interventions, all of which are potentially associated with transient bacteraemia.

In detail, Rutterford et al. evaluated the administration of amoxicillin (2–3 g) or other beta-lactam antibiotics 30–60 minutes before the procedure compared with a placebo or no prophylaxis in regard to patients with ha high risk of developing IE. The only study included in the review was one by Van der Meer et al. from 1992 [116], which found that prophylaxis did not result in a statistically significant reduction in the risk of endocarditis among patients undergoing scaling or similar treatments (OR 1.62; 95% CI 0.57–4.57). However, a detailed analysis of associated procedures revealed that 13% of IE cases had undergone subgingival scaling, and 25% had received scaling combined with polishing (root surface debridement), and, notably, 43% of the control patients underwent the same procedures without developing IE. These findings, while not conclusive, highlight a possible association between such interventions and bacteraemia, even in the absence of demonstrated efficacy of antibiotic prophylaxis in preventing endocarditis in this context.

It is also important to note that oral amoxicillin combined with clavulanic acid has not shown any additional benefit in this setting [116].

In patients with penicillin allergy, oral azithromycin appears to be more effective than clindamycin in reducing bacteraemia, as reported by Lafaurie et al. (2019) [127]. Specifically, Lafaurie et al. documented that pre-procedural use of amoxicillin, azithromycin, and clindamycin (as per AHA protocols) led to reductions in bacteraemia risk of 59%, 49%, and 11%, respectively, compared to no prophylaxis. Aggregated estimates also showed that two antibiotics not included in AHA protocols—namely, moxifloxacin and intravenous amoxicillin–clavulanic acid—were associated with significant reductions in bacteraemia following invasive dental procedures amounting to 41% and 99%, respectively [127].

### 4.6. Future Research Perspectives

A key area of research, still widely debated yet presenting notable gaps and uncertainties, is antibiotic prophylaxis and how it may reduce the risk of endocarditis in patients with periodontal disease—particularly in the context of evaluating the effectiveness of antibiotic regimens and resistance patterns based on the specific types of oral bacteria involved. Optimising antibiotic prophylaxis for high-risk patients undergoing invasive dental procedures should play a crucial role in public health, reducing both complications and antibiotic resistance.

Innovative directions for future studies could include the development of personalised prophylactic strategies based on individual risk profiles, taking into account comorbidities, the composition of the oral microbiota, and genetic predisposition [207]. This would allow for the design of targeted antibiotic protocols rather than universal approaches, thereby minimising both the risk of infective endocarditis and the emergence of antibiotic resistance [208].

Another promising line of investigation concerns the role of epigenetic and inflammatory markers as predictors of systemic complications related to oral infections [209]. Longitudinal studies evaluating these markers could help identify patients at higher risk of cardiovascular events and guide early intervention strategies [210].

Moreover, more attention should be given to specific high-risk populations, such as children with congenital heart disease, elderly patients with comorbidities, immunocompromised individuals, and patients with “intermediate-risk” cardiac conditions (e.g., bicuspid aortic valve and mitral valve prolapse), whose classification may need to be revised based on recent evidence.

Additionally, the implementation of real-world data platforms [211] and multi-omics approaches [212] could enhance our understanding of the oral–systemic interface and the impact of oral health interventions on cardiovascular outcomes [212].

Finally, public health strategies and educational programs targeted at preventing oral diseases in at-risk cardiac patients should be developed and evaluated [213]. These may include oral hygiene campaigns, antimicrobial stewardship programs in dentistry, and interdisciplinary clinical pathways involving both cardiologists and dental professionals [214].

Recent advances in artificial intelligence (AI) provide innovative opportunities to enhance adherence to antibiotic prophylaxis protocols for the prevention of IE. Among these innovations, the implementation of AI tools such as Large Language Models (LLMs) has gained particular attention due to their efficiency and accessibility, facilitating their adoption by dental healthcare professionals. In this context, a study conducted by Rewthamrongsris et al. in 2025 [215] assessed the accuracy of seven recent LLMs in recommending antibiotic prophylaxis for dental procedures as a preventive strategy against IE. These models may serve as accessible instruments for rapidly retrieving detailed information and clinical protocols for physicians and dentists.

The cited study revealed that the evaluated LLMs achieved an accuracy ranging from 68.57% to 80%, indicating that these models are not yet optimally trained to address highly specific questions within the medical–dental domain. Additionally, inconsistencies, difficulties in managing complex question structures, and occasional contradictions were observed. The authors concluded that although LLMs exhibit promising potential, they should currently be regarded as supplementary tools that necessitate careful evaluation and verification by healthcare professionals, particularly in complex clinical decision-making contexts.

Future research should focus on the domain-specific training of LLMs, their integration with electronic health records, and the development of AI-assisted clinical decision support systems specifically tailored to dental care and the prevention of infective endocarditis [216]. Such advancements may contribute to reducing variability in antibiotic-prescribing practices and to promoting evidence-based clinical approaches [215].

### 4.7. Limitations of the Review

As this review focused exclusively on systematic reviews, a considerable number of original studies were excluded, which may have limited the variety of evidence considered. Observational, clinical, or cohort studies that could have provided complementary information were excluded.

The data extraction process primarily focused on the main outcomes of the systematic reviews. However, potential secondary or more detailed outcomes may not have been included, limiting the depth of the analysis.

The search was conducted using three main databases (PubMed, Scopus, and ScienceDirect) and Google Scholar for grey literature. However, the exclusive use of these tools may have excluded studies published in other journals or narrower databases, which could contain relevant evidence.

## 5. Conclusions

This mapping review examined the existing literature on the correlation between cardiovascular diseases, particularly endocarditis, and oral disorders, with a specific focus on periodontitis and dental caries. The results suggest that there is a significant connection between oral health and cardiovascular health, highlighting how oral infections, particularly periodontitis, can contribute to bacteraemia and promote the development of cardiovascular complications, including infective endocarditis.

Antibiotic prophylaxis for preventing infective endocarditis, especially in high-risk patients, remains a controversial topic. While some systematic reviews have highlighted the effectiveness of prophylaxis in specific contexts, its universal applicability is still under discussion, with studies suggesting the need for further research to clarify its real effectiveness.

## Figures and Tables

**Figure 1 dentistry-13-00215-f001:**
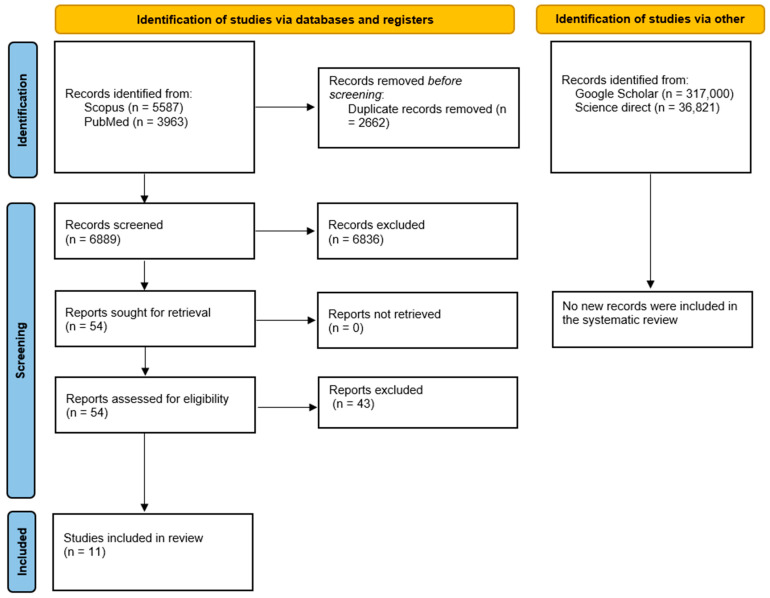
Flow chart for the mapping review.

**Table 1 dentistry-13-00215-t001:** Excluded studies.

Author, Data, Reference	Main Topics	Reasons for Exclusion
**Dave and Tattar, 2025** [15]	Bacterial antimicrobial resistance	A
**Zorman et al., 2025** [16]	Incidence of bioprosthetic mitral valve thrombosis	A
**Barda et al., 2024** [17]	Intravenous treatment for infective endocarditis	A
**Sonaglioni et al., 2024** [18]	Echocardiographic assessment of mitral valve prolapse	A
**Mourad et al., 2025** [19]	Oral versus intravenous antibiotic therapy for *Staphylococcus aureus* bacteraemia or endocarditis	A
**Veldurthy et al., 2025** [20]	Incidence of infective endocarditis following transcatheter pulmonary valve replacement	A
**Martins et al., 2024** [21]	Bacteraemia following different oral procedures	A
**Monaci et al., 2024** [22]	Infective endocarditis appearing subsequent to transcatheter versus surgical aortic valve replacement	A
**Brown et al., 2024** [23]	Treatment of endocarditis for people who inject drugs	A
**Barbosa-Ribeiro et al., 2024** [24]	Bacterial antimicrobial resistance	A
**Nor et al., 2023** [25]	Systemic lupus erythematosus and cardiovascular diseases	A
**Diz Dios et al., 2023** [26]	Prophylaxis guidelines for patients with high-risk cardiac conditions	D
**Lean et al., 2023** [27]	Prophylactic antibiotic use for infective endocarditis	A
**Budea et al., 2023** [28]	Clinical characteristics of infective endocarditis among older adults and the risk factors that could lead to adverse outcomes	A
**Araújo Júnior et al., 2022** [29]	Presence of amoxicillin-resistant streptococci in the mouths of children	A
**Hussein et al., 2022** [30]	Quality appraisal of antibiotic prophylaxis guidelines	D
**Shimizu et al., 2022** [31]	*Parvimonas micra* infection route	B
**Alifragki et al., 2022** [32]	Infective endocarditis caused by *Pasteurella* Species	B
**Sindon et al., 2022** [33]	Health risks posed by body piercings	A
**Talha et al., 2021** [34]	Temporal trends of infective endocarditis	A
**Wald-Dickler et al., 2022** [35]	Comparison of intravenous and oral regimens	A
**Franconieri et al., 2021** [36]	*Rothia* spp.-related infective endocarditis	B
**Herrera-Hidalgo et al., 2020** [37]	Endocarditis caused by *Enterococcus faecalis*	A
**Martí-Carvajal et al., 2020** [38]	A comparison of different antibiotic regimens	A
**Cummins et al., 2020** [39]	Dentists’ knowledge of the relevant guidelines for prescribing antibiotic prophylaxis for the prevention of infective endocarditis	D
**Vahabi et al., 2019** [40]	Review of the infective endocarditis literature published in Turkey	A
**Khan et al., 2020** [41]	Infective endocarditis acquired post-transcatheter-aortic-valve implantation	A
**Napolitani et al., 2019** [42]	*Kocuria kristinae*	A
**Nakatani et al., 2019** [43]	Guidelines on prevention and treatment of infective endocarditis	D
**Abarbanell et al., 2019** [44]	Contraceptive use among women with congenital heart disease	A
**Singh Gill et al., 2018** [45]	Antibiotic prophylaxis in relation to dental implants and extraction procedures	B
**Russell et al., 2018** [46]	Rheumatic heart disease	A
**Al-Omari et al., 2014** [47]	The role of oral antibiotic therapy in treating infective endocarditis	A
**Glenny et al., 2013** [48]	Prophylaxis of bacterial endocarditis in dentistry	C
**Esposito et al., 2013** [49]	Antibiotics employed during dental implant placement	A
**Tepper et al., 2010** [50]	Safety of contraceptives	A
**Swedish Council on Health Technology, 2010** [51]	Antibiotic prophylaxis for surgical procedures	D
**Esposito et al., 2010** [52]	Antibiotics used during dental implant placement	C
**Oliver et al., 2008** [53]	Prophylaxis of bacterial endocarditis in dentistry	C
**Esposito et al., 2008** [54]	Antibiotics used during dental implant placement	C
**Lockhart et al., 2007** [55]	Efficacy of antibiotic prophylaxis in dental practice	B
**Oliver et al., 2004** [56]	Prophylaxis of bacterial endocarditis in dentistry	C
**Esposito et al., 2003** [57]	Antibiotics used during dental implant placement	C

(A) Studies that did not specifically address the relationship between endocarditis and oral disease but instead discussed broader cardiovascular implications or unrelated infections; (B) studies that lacked sufficient methodological transparency (i.e., studies whose protocols were not registered or research associated with a lack of well-defined protocol or unclear inclusion criteria); (C) studies that were duplicate systematic reviews or had substantial overlaps with previously published reviews and thus constitute an update; (D) studies that were not systematic reviews but clinical guidelines or position papers or questionnaires.

**Table 2 dentistry-13-00215-t002:** Data extraction table showing the information regarding the papers and the main results of the systematic reviews. Not all included studies provided data for the meta-analyses.

Autor, Data, Reference	Country	Outcome	Outcome from Results	Studies Included		Meta-Analysis	PROTOCOL	Risk of Bias
**Kussainova et al., 2025** [58]	Kazakhstan, China, Switzerland	IE in invasive dental procedures	IE and invasive dental procedures (OR 1.49, 95% CI 1.25–1.76);	9	Lacassin et al., 1995 [59] Strom et al., 1998 [60] Porat et al., 2008 [61] Chen et al., 2015 [62] Chen et al., 2018 [63] Tubiana et al., 2017 [64] Thornhill et al., 2023 [65] Thornhil et al., 2022 [66] Thornhill et al., 2024 [67]	yes	PROSPERO: CRD42023488546	Robins
**Sperotto et al., 2024** [68]	USA, South Africa, Portugal, Singapore, Italy, UK, Spain	AP	AP was associated with a lower risk of IE after invasive dental procedures in individuals with high risk (RR, 0.41; 95% CI, 0.29–0.57;	30	Keller et al., 2017 [69] van den Brink et al., 2017 [70] Bates et al., 2017 [71] Sakai Bizmark et al., 2017 [72] Garg et al., 2019 [73] Quan et al., 2020 [74] Vähäsarja et al., 2020 [75] Bikdeli et al., 2013 [76] DeSimone et al., 2015 [77] Toyoda et al., 2017[78] Chen et al., 2015 [62] Sun et al., 2017 [79] Chen et al., 2018 [63] Thornhil et al., 2022 [66] Tubiana et al., 2017 [64] Thornhill et al., 2022 [80] Thornhill et al., 2023 [65] Thornhill et al., 2024 [67] Rogers et al., 2008 [81] Pasquali et al., 2015 [82] Pant et al., 2015 [83] Thornhill et al., 2017 [84] DeSimone et al., 2021 [85] Mackie et al., 2016 [86] Knirsch et al., 2020 [87] Weber et al., 2022 [88] Krul et al., 2015 [89] Duval et al., 2012 [90] Dayer et al., 2015 [91] Shah et al., 2020 [92]	yes	PROSPERO, CRD4202017398	EPOC, NHLBI
**Friedlander and Couto-Souza, 2023** [93]	USA, Brazil	AP	Significant risk of developing IE	3	van den Brink et al., 2017 [70] Zegri-Reiriz et al., 2018 [94] Thornhill et al., 2018 [95]	no	no	no
**Albakri et al., 2022** [96]	Germany, India	AP, Risk of bacteraemia	AI RR: 0.51; 95% CI; 0.45–0.58).	17	Khairat, 1966 [97] Shanson et al., 1985 [98] Maskell et al., 1986 [99] Shanson et al., 1987 [100] Vergis et al., 2001 [101] Lockhart et al., 2004 [102] Diz Dios et al., 2006 [103] Abu-Ta’a et al., 2008 [104] Anitua et al., 2008 [105] Lockhart et al., 2008 [106] Asi et al., 2010 [107] Esposito et al., 2010 [108] Siddiqi et al., 2010 [109] Chandramohan et al., 2011 [110] Maharaj et al., 2012 [111] DuVall et al., 2013 [112] Limeres Posse et al., 2016 [113]	yes	INPLASY: INPLASY202270011	no
**Bergadà-Pijuan et al., 2023** [114]	Switzerland	Incidence of IE following only invasive dental procedures	RR 0.39, *p*-value 0.11).	1	Tubiana et al., 2017 [64]	yes	PROSPERO: CRD42020175398	ok
**Rutherford et al., 2022** [115]	UK	AP	No clear evidence	1	Van der Meer et al., 1992 [116]	no	Cochrane Oral Health Trials Register	Cochrane revised tool
**Karikoski et al., 2021** [117]	Finland	Prevalence of caries in children with congenital heart disease	Higher caries prevalence compared with healthy controls	9	Pourmoghaddas et al., 2018 [118] Ali et al., 2017[119] Cantekin et al., 2015 [120] Cantekin et al., 2013 [121] Suma et al., 2011[122] Siahi-Benlarbi et al., 2010 [123] da Fonseca et al., 2009 [124] Tasioula et al., 2008 [125] Stecksén-Blicks et al., 2004 [126]	no	NO	Newcastle–Ottawa scale for cross-sectional studies
**Lafaurie et al., 2019** [127]	Colombia	AP	Antibiotics significantly reduced the bacteraemia RR, 0.50; 95% CI 0.38–0.67	12	Snanson et al., 1978 [128] Roberts et al., 1987 [129] Hall et al., 1993 [130] Hall et al., 1996 [131] Wahlmann et al., 1999 [132] Vergis et al., 2001 [101] Lockhart et al., 2004 [102] Diz Dios et al., 2006 [103] Lockhart et al., 2008 [106] Maharaj et al., 2012 [111] DuVall et al., 2013 [112] Limeres Posse et al., 2016 [113]	Yes	PROSPERO: CRD42018085836	Cochrane revised tool
**Lockhart et al., 2019** [133]	USA	Dental treatment before cardiac valve surgery (RR)	Mortality, the pooled RR of all-cause mortality was 1.00, 95% CI, 0.53–1.91	6	de Souza et al., 2016 [134] Deppe et al., 2007 [135] Wu et al., 2008 [136] Hakeberg et al., 1999 [137] Nakamura et al., 2011 [138] Bratel et al., 2011 [139]	yes	PROSPERO: CRD42018090110	Cochrane risk-of-bias tool, Newcastle–Ottawa
**González Navarro et al., 2017** [140]	Spain, Portugal	AP	AI before an invasive dental procedure does not prevent bacteraemia	32	Lockhart eta al., 2009 [141] Bahrani-Mougeot et al., 2008 [142] Maharaj et al., 2012 [111] Lockhart et al., 2008 [106] Heimdahl et al., 1980[143] Hall et al., 1993 [130] Lockhart et al., 2004 [102] Rajasuo et al., 2004 [144] Diz Dios et al., 2006 [103] Roberts et al., 2006 [145] Tomás et al., 2007 [146] Roberts et al., 1998 [147] Benítez-Páez et al., 2013 [148] Maharaj et al., 2012 [149] Peterson et al., 1976 [150] Rahn et al., 1995 [151] Maskell et al., 1986 [99] Sefton et al., 1990 [152] Hall et al., 1996 [153] Hall et al., 1996 [131] Tuna et al., 2012 [154] Sweet et al., 1978 [155] Shanson et al., 1985 [98] Shanson et al., 1987 [100] Cannell et al., 1991 [156] Aitken et al., 1995 [157] Wahlmann et al., 1999 [132] Vergis et al., 2001 [101] Josefsson et al., 1985 [158] Tomás et al., 2008 [159] Duvall et al., 2013 [112] Piñeiro et al., 2010 [160]	no	no	no
**Cahill et al., 2017** [161]	UK	AP, Risk of bacteraemia	RR 0.53, 95% CI 0.49–0.57, OR 0.59; 95%, CI 0.27–1.30; *p* = 0.14	37	Imperiale and Horwitz, 1990 [162] Lacassin et al., 1995 [59] Van der Meer et al., 1992 [116] Asi et al., 2010 [107] Baltch et al., 1982 [163] Cannell et al., 1991 [156] Coulter, et al., 1990 [164] Diz Dios et al., 2006 [103] DuVall et al., 2013 [112] Hall et al., 1993 [130] Hall et al., 1996 [131] Head et al., 1984 [165] Khairat, 1966 [97] Limeres Posse et al., 2016 [113] Lockhart et al., 2008 [106] Lockhart et al., 2004 [102] Maharaj et al., 2012 [111] Maskell et al., 1986 [99] Roberts et al., 1987 [129] Shanson et al., 1985 [98] Shanson et al., 1987 [100] Shanson et al., 1978 [128] Vergis et al., 2001 [101] Horstkotte et al., 1987 [166] Strom et al., 1998 [60] Bates et al., 2017 [71] Bikdeli et al., 2013 [76] Dayer et al., 2015 [91] Thornhill et al., 2011 [167] DeSimone et al., 2015 [77] Desimone et al., 2012 [168] Duval et al., 2012 [90] Keller et al., 2017 [69] Mackie et al., 2016 [86] Pant et al., 2015 [83] Salam et al., 2014 [169] van den Brink et al., 2017 [70]	yes	no	Cochrane risk-of-bias tool

RR (risk ratio), OR (odds ratio), IE (infective endocarditis), AP (antibiotic prophylaxis), and CI (confidence interval).

**Table 3 dentistry-13-00215-t003:** Summary of results from the included systematic reviews on infective endocarditis, oral conditions, and antibiotic prophylaxis.

Autor Data Reference	
**Kussainova et al., 2025** [58]	In 2025, Kussainova et al. found an association between IE and invasive dental procedures (OR 1.49, 95% CI 1.25–1.76; *p* < 0.00001), while subgroup analysis showed an increased risk of IE following tooth extraction (OR 2.73, 95% CI 1.46–5.11; *p* = 0.002) and oral surgery (OR 6.33, 95% CI 2.43–16.49; *p* = 0.0002) for high-risk patients. The strongest association they identified was between IE and tooth extraction (OR 1.90, 95% CI 1.17–3.08; *p* = 0.010; I^2^ = 80%;) and oral surgery (OR 3.11, 95% CI 1.20–8.05; *p* = 0.02; I^2^ = 77%;), while there was no significant association between IE and invasive dental procedures such as scaling (OR 1.00, 95% CI 0.85–1.18; *p* = 1.00; I^2^ = 0%;), endodontic treatment (OR 1.04, 95% CI 0.73–1.49; *p* = 0.82; I^2^ = 0%;), and periodontal treatment (OR 0.69, 95% CI 0.28–1.67; *p* = 0.41; I^2^ = 69%;).
**Sperotto et al., 2024** [68]	In 2024, Sperotto et al. found that antibiotic prophylaxis was associated with a significantly lower risk of infective endocarditis after invasive dental procedures among high-risk individuals (pooled RR, 0.41; 95% CI, 0.29–0.57; P for heterogeneity = 0.51; I^2^, 0%).
**Friedlander and Couto-Souza, 2023** [93]	In 2023, Friedlander and Couto-Souza sought to investigate whether the European Society of Cardiology (ESC) guidelines, which recommend AP only for “high-risk” patients, are also appropriate for patients with valvular heart disease defined as “intermediate risk” (such as bicuspid aortic valve—BAV—or mitral valve prolapse—MVP). They found that patients with BAV or MVP, traditionally classified as “intermediate risk” according to current ESC guidelines, may actually be at significantly greater risk of developing infective endocarditis (IE) and related complications. Based on large-scale data from the Netherlands, Spain, and the UK, the authors reported a higher incidence of IE, hospital mortality, and intracardiac complications in these patients, supporting a reclassification of BAV and MVP as “high-risk” conditions requiring prophylactic antibiotic coverage before high-risk dental procedures.
**Albakri et al., 2022** [96]	Albakri et al. (2022) [96] compared AP with a placebo. The results showed that AP significantly reduced the incidence of bacteraemia by 49% (risk ratio 0.51; 95% CI: 0.45–0.58; *p* = 0.0001). Although bacteraemia has been used as a surrogate marker for IE, the authors acknowledged that direct evidence linking AP to a reduction in IE remains inconclusive due to the rarity of the disease and the impracticality of conducting large-scale studies.
**Bergadà-Pijuan et al., 2023** [114]	Bergadà-Pijuan et al. (2023) [114] conducted a systematic review to assess the efficacy of AP for IE for adults undergoing dental procedures. Of the 264 publications reviewed, only one prospective cohort study (Tubiana et al., 2017) [64] met the inclusion criteria. This study focused exclusively on high-risk patients with prosthetic heart valves. The results showed a non-significant reduction in the incidence of IE when AP was administered (RR 0.39; *p* = 0.11), suggesting a potential protective effect, although the results are inconclusive. The authors concluded that the evidence for or against AP use remains weak, particularly for low- and moderate-risk populations, and highlighted the urgent need for well-powered clinical trials to clarify current guideline recommendations.
**Rutherford et al., 2022** [115]	In the systematic review conducted by Rutherford et al. (2022) [115], the authors aimed to assess whether AP before invasive dental procedures reduces the incidence of bacterial endocarditis in at-risk individuals. Despite extensive database searches, only one eligible case–control study was included (Van der Meer et al., 1992) [116]. This study found no statistically significant difference in the incidence of bacterial endocarditis between patients who received penicillin prophylaxis and those who did not (OR 1.62; 95% CI: 0.57–4.57). The overall certainty of the evidence was judged to be very low, and no data were available regarding mortality, adverse events, or cost-effectiveness. The authors concluded that there is still no reliable evidence to support or refute the effectiveness of AP in preventing BE after dental procedures in high-risk populations and that clinical decisions should be guided by shared decision making and discussion of potential benefits and risks.
**Karikoski et al., 2021** [117]	Karikoski et al. (2021) [117] conducted a systematic review comparing the prevalence of dental caries in children suffering from congenital heart disease (CHD) with that in healthy peers. Of the nine included studies, seven reported a higher prevalence of caries in children with CHD, with statistically significant differences reported in three. The mean dmft and DMFT scores were consistently higher in the CHD group. For example, in one study, 77.4% of children with CHD had caries in their primary teeth (dmft > 0); in comparison, this figure was 56.5% for the controls. Evidence suggests that children with complex or surgically treated CHD (severity grades 2–4) are at particular risk. Despite heterogeneity and limitations in study quality and design, the review concluded that children with CHD have a higher caries burden, highlighting the need for targeted prevention strategies in this vulnerable population.
**Lafaurie et al., 2019** [127]	Lafaurie et al. (2019) [127] conducted a review to evaluate the effectiveness of AP in reducing bacteraemia after tooth extractions. Pooled results showed that AP reduced the incidence of bacteraemia by 50% (RR 0.50; 95% CI: 0.38–0.67). Amoxicillin, azithromycin, and clindamycin (AHA protocol) demonstrated variable efficacy, with an intravenous amoxicillin–clavulanic acid combination achieving the greatest reduction (RR 0.01).
**Lockhart et al., 2019** [133]	Lockhart et al. (2019) [133] conducted a review to assess whether professional dental treatment before heart valve surgery (CVS) or LVAD implantation reduces postoperative complications. The results showed no statistically significant differences in key outcomes, including all-cause mortality (RR 1.00; 95% CI: 0.53–1.91), infective endocarditis (RR 1.30; 95% CI: 0.51–3.35), postoperative infections (RR 1.01; 95% CI: 0.76–1.33), and length of hospital stay (mean difference: +2.9 days; 95% CI: −2.3 to 8.1). The certainty of the evidence was assessed to be very low due to risk of bias and imprecision. The authors concluded that current data do not support a definitive benefit or harm of preoperative dental treatment for adults undergoing CVS, emphasising the need for individualised, interdisciplinary decision making.
**González Navarro et al., 2017** [140]	González Navarro et al. (2017) [140] evaluated the duration and extent of bacteraemia after oral surgery and the potential impact of AP. Their review confirmed that although AP does not completely prevent post-procedure bacteraemia, it significantly reduces its extent and persistence. Amoxicillin was the most frequently studied antibiotic, followed by clindamycin, erythromycin, teicoplanin, and others. The most commonly isolated microorganism was *Viridans streptococcus*. Despite the variability of the results, high-dose systemic amoxicillin administration (e.g., 2–3 g 1 h before surgery) was associated with the most consistent protective effect. However, several studies reported no significant benefit compared to a placebo. The authors concluded that although AP may reduce the risk of IE in high-risk subjects, standardised protocols and further high-quality studies are needed to clarify the clinical utility of AP in dental surgery.
**Cahill et al., 2017** [161]	Cahill et al. (2017) [161] conducted a review to assess the effectiveness of AP in preventing IE following dental procedures. While AP significantly reduced the incidence of post-procedure bacteraemia (RR 0.53; 95% CI: 0.49–0.57), the evidence supporting a protective effect against IE was inconclusive. Observational studies showed a non-significant trend in favour of AP (OR 0.59; 95% CI: 0.27–1.30; *p* = 0.14) and only one time-trend study (from the UK) reported an increase in the incidence of IE following complete cessation of AP. The authors concluded that the current evidence base is limited by methodological heterogeneity and a lack of randomised clinical trials but recommended that PA may remain a justified low-risk intervention for high-risk individuals, in accordance with existing ESC and ACC/AHA guidelines.

**Table 4 dentistry-13-00215-t004:** Risk of bias.

First Author, Data	Phase 1	Phase 2				Phase 3
	PICO	Study Eligibility Criteria	Identification and Selection of Studies	Data Collection and Study Appraisal	Synthesis and Findings	Risk of Bias in the Review
Kussainova et al., 2025 [58]	OK	OK	OK	OK	OK	OK
Sperotto et al., 2024 [68]	OK	OK	?	OK	OK	OK
Friedlander and Couto-Souza, 2023 [93]		?	?	?	OK	OK
Albakri et al., 2022 [96]	OK	OK	OK	?	OK	OK
Bergadà-Pijuan et al., 2023 [114]	OK	OK	OK	OK	OK	OK
Rutherford et al., 2022 [115]	OK	OK	OK	OK	OK	OK
Karikoski et al., 2021 [117]	OK	OK	?	OK	OK	OK
Lafaurie et al., 2019 [127]	OK	OK	OK	OK	OK	OK
Lockhart et al., 2019 [133]	OK	OK	OK	OK	OK	OK
González Navarro et al., 2017 [140]	OK	OK	?	?	OK	OK
Cahill et al., 2017 [161]	OK	OK	?	?	OK	OK

ROBIS scale: OK (low); ? (unclear).

## Data Availability

No new data were created.

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
