# Peer review of "Mapping Review of the Correlations Between Periodontitis, Dental Caries, and Endocarditis"

_dentistry, 2025, doi:10.3390/dj13050215_

Round 1
Reviewer 1 Report
Comments and Suggestions for Authors
Dear Authors,
I read your mapping review on the correlation between cardiovascular pathology and oral health with interest.
I enclose my revisions.
Introduction:
- The objective of the study should be relocated to the conclusion of the introduction.
- Incorporate correlation mechanisms to introduce the topic, even if these are subsequently addressed in the discussion.
- Please consider adding or replacing with the following reference, which is deemed more appropriate: Nocini R et al. Periodontitis, coronary heart disease and myocardial infarction: treat one, benefit all. Blood Coagul Fibrinolysis. 2020 Sep;31(6):339-345.
Methods:
- Please include the exclusion criteria and specify whether filters were utilized in the search process.
Results:
- Provide a brief description of the included studies, highlighting the main characteristics and evaluated outcomes.
Discussion:
- The section on the correlation with caries appears to be missing and should be included to complete the discussion.
In line 297, within the evidence of literature, please add the following reference from the working group of Pardo A. et al: Detection of Periodontal Pathogens in Oral Samples and Cardiac Specimens in Patients Undergoing Aortic Valve Replacement: A Pilot Study. J Clin Med., this reference discusses the correlation of periodontitis with aortic valve disease, incorporating a microbiological perspective.
Conclusion: The conclusion is satisfactory.
Author Response
ANSWER
We sincerely thank the reviewer for carefully reading our manuscript and for the insightful and constructive comments. The answers are highlighted in green, the parts added to the manuscript are highlighted in yellow
- The objective of the study should be relocated to the conclusion of the introduction.
The focus of the review has been moved to the end of the introduction
- Incorporate correlation mechanisms to introduce the topic, even if these are subsequently addressed in the discussion.
- the part concerning the correlation mechanisms to introduce the topic has been incorporated, as requested
Several mechanisms have been proposed to explain the correlation between oral infections and cardiovascular conditions such as endocarditis. Among these, chronic inflammation caused by periodontitis or caries may induce systemic inflammatory re-sponses, promoting endothelial dysfunction and creating a favorable environment for microbial colonization. Furthermore, repeated episodes of transient bactere-mia—triggered by daily oral activities or invasive dental procedures—enable oral bac-teria, particularly Streptococcus species, to reach and adhere to damaged cardiac tis-sues. The virulence factors of these bacteria, combined with an impaired immune re-sponse or pre-existing cardiac lesions, may facilitate the formation of vegetations on heart valves, ultimately leading to infective endocarditis. These mechanisms underline the importance of exploring the pathogenic interplay between oral and cardiovascular health, as will be discussed in detail in the following sections[1].
- Please consider adding or replacing with the following reference, which is deemed more appropriate: Nocini R et al. Periodontitis, coronary heart disease and myocardial infarction: treat one, benefit all. Blood Coagul Fibrinolysis. 2020 Sep;31(6):339-345.
- the reference has been added as requested
Methods:
- Please include the exclusion criteria and specify whether filters were utilized in the search process.
- Exclusion criteria were added and any filters were specified in the search process as required. The following parts were added to the manuscript
The exclusion criteria applied included: (i) narrative reviews, editorials, letters to the editor and case reports; (ii) studies that did not address the association between oral diseases and endocarditis or cardiovascular diseases; (iii) duplicate publications; (iv) reviews lacking sufficient methodological transparency or a clear description of the included studies. Only systematic reviews were considered for inclusion in order to ensure a high-quality synthesis of evidence.
During the search process, filters were applied where available to refine the results and include review articles. In PubMed, the filter "Systematic review" was selected to prioritize high-quality evidence. No language restrictions were applied, as long as an abstract was available in English. The search was limited to human studies.
Discussion:
- The section on the correlation with caries appears to be missing and should be included to complete the discussion.
The section on caries correlation has been added as requested.
4.3. Correlation with Dental Caries
Although periodontitis has been studied more extensively in relation to infective endocarditis[2], the potential role of dental caries and its complications must be taken into consideration. Dental caries, especially if untreated, can progress to pulpitis and subsequently to apical periodontitis, both associated with local inflammatory responses with possible systemic implications[3]. These infections can lead to abscess formation and contribute to transient or prolonged episodes of bacteremia [4], particularly with the involvement of pathogens such as Streptococcus mutans [5], Lactobacillus spp.[6] and Enterococcus faecalis [7], which have been implicated in cases of infective endocarditis.
Some epidemiological studies have identified a higher prevalence of dental caries and untreated pulp infections in patients with congenital heart disease and a history of endocarditis [8]. The caries process itself may not directly cause endocarditis, but it creates a portal of entry for bacteria into the bloodstream, especially when accompanied by inadequate oral hygiene or during procedures such as endodontic therapy [9] and tooth extractions [4]. Furthermore, caries-related pulp necrosis often requires surgical intervention, increasing the risk of bacteremia in susceptible individuals.
Therefore, although the evidence linking dental caries to endocarditis is less robust than for periodontitis, the literature suggests that untreated caries and secondary infections should be considered significant risk factors, particularly in high-risk cardiac patients. This highlights the importance of early diagnosis, prevention and management of caries as part of cardiovascular risk reduction strategies.
In line 297, within the evidence of literature, please add the following reference from the working group of Pardo A. et al: Detection of Periodontal Pathogens in Oral Samples and Cardiac Specimens in Patients Undergoing Aortic Valve Replacement: A Pilot Study. J Clin Med., this reference discusses the correlation of periodontitis with aortic valve disease, incorporating a microbiological perspective.
- the reference has been added as requested

Reviewer 2 Report
Comments and Suggestions for Authors
This study conducts a rigorous and standardized analysis of the correlation between systemic diseases, particularly cardiovascular conditions, and the most prevalent odontological pathologies. It underscores the critical importance of implementing evidence-based antibiotic prophylaxis in high-risk patients and provides valuable perspectives for future clinical guidelines. However, the subject is extensively documented in the existing literature, with a considerable body of research already addressing this issue.
The literature review is meticulously structured and methodologically rigorous, integrating data from the most recent scientific publications. However, given the extensive prior exploration of this subject, its novelty and impact may be limited for the reader.
I recommend enhancing the study’s scientific appeal by incorporating more innovative research directions. This would provide greater novelty and relevance, particularly considering that the study ultimately reinforces the importance of oral health in preventing cardiovascular complications, especially infective endocarditis.
Comments on the Quality of English LanguageEnglish could be improved.
Author Response
Answer
We sincerely thank the Reviewer for their positive evaluation and constructive suggestions. In response to the valuable comment, we have expanded the “Future Research Perspectives” section to include innovative directions, such as the integration of artificial intelligence (AI) tools like Large Language Models (LLMs) for supporting antibiotic prophylaxis decision-making including a chapter on the use and accuracy of Large model language in Endocarditis Prophylaxis in Dental Procedures. the chapter Future Research Perspectives has been rewritten.
The answers are highlighted in green, the parts added to the manuscript are highlighted in yellow
Future Research Perspectives
A key area of research, still widely debated yet presenting notable gaps and uncertainties, is antibiotic prophylaxis and how it may reduce the risk of endocarditis in patients with periodontal disease—particularly in the context of evaluating the effectiveness of antibiotic regimens and resistance patterns based on the specific types of oral bacteria involved. Optimizing antibiotic prophylaxis for high-risk patients undergoing invasive dental procedures should play a crucial role in public health, reducing both complications and antibiotic resistance.
Innovative directions for future studies could include the development of personalized prophylactic strategies based on individual risk profiles, taking into account comorbidities, the composition of the oral microbiota, and genetic predisposition [206]. This would allow for the design of targeted antibiotic protocols rather than universal approaches, thereby minimizing both the risk of infective endocarditis and the emergence of antibiotic resistance [207].
Another promising line of investigation concerns the role of epigenetic and inflammatory markers as predictors of systemic complications related to oral infections [208]. Longitudinal studies evaluating these markers could help identify patients at higher risk of cardiovascular events and guide early intervention strategies [209].
Moreover, more attention should be given to specific high-risk populations, such as: children with congenital heart disease, elderly patients with comorbidities, immunocompromised individuals, and patients with “intermediate-risk” cardiac conditions (e.g., bicuspid aortic valve and mitral valve prolapse), whose classification may need to be revised based on recent evidence.
Additionally, the implementation of real-world data platforms [210] and multi-omics approaches [211] could enhance our understanding of the oral-systemic interface and the impact of oral health interventions on cardiovascular outcomes [211].
Finally, public health strategies and educational programs targeted at preventing oral diseases in at-risk cardiac patients should be developed and evaluated [212]. These may include oral hygiene campaigns, antimicrobial stewardship programs in dentistry, and interdisciplinary clinical pathways involving both cardiologists and dental professionals [213].
Recent advances in artificial intelligence (AI) provide innovative opportunities to enhance adherence to antibiotic prophylaxis protocols for the prevention of IE. Among these innovations, the implementation of AI tools such as Large Language Models (LLMs) has gained particular attention due to their efficiency and accessibility, facilitating their adoption by dental healthcare professionals. In this context, a study conducted by Rewthamrongsris et al. in 2025 [214]assessed the accuracy of seven recent LLMs in recommending antibiotic prophylaxis for dental procedures as a preventive strategy against IE. These models may serve as accessible instruments for rapidly retrieving detailed information and clinical protocols for physicians and dentists.
The study revealed that the evaluated LLMs achieved an accuracy ranging from 68.57% to 80%, indicating that these models are not yet optimally trained to address highly specific questions within the medical-dental domain. Additionally, inconsistencies, difficulties in managing complex question structures, and occasional contradictions were observed. The authors concluded that, although LLMs exhibit promising potential, they should currently be regarded as supplementary tools that necessitate careful evaluation and verification by healthcare professionals, particularly in complex clinical decision-making contexts.
Future research should focus on domain-specific training of LLMs, their integration with electronic health records, and the development of AI-assisted clinical decision support systems specifically tailored for dental care and the prevention of infective endocarditis [215]. Such advancements may contribute to reducing variability in antibiotic prescribing practices and to promoting evidence-based clinical approaches[214].

Reviewer 3 Report
Comments and Suggestions for Authors
Thank you for the opportunity to review your review on endocarditis periodontitis and decay.
Your methods to find relevant scientific reviews are appropriate but needs more details for the exlusion of the last 40 papers.
How many studies were included by the included reviews? Propably some studies were included in several of them? (BIAS)
The outcomes you looked at should be clearly appointed. The results of the outcomes could be presented in a separate figure or table. Preventive antibiotic usage is an outcome in only 4/5 included papers. Please dicuss the different antibiotic regimes more in detail. What evidence did you find on the use of antibiotics for caries and periodontal treatment and prevention of endocarditis? Did you find any differences in studies included where severe periodontitis cases (stage 3+4) were treated? Did you find any age dependency in the includes studies?

Author Response
ANSWER
We sincerely thank the reviewer for carefully reading our manuscript and for the insightful and constructive comments .The answers are highlighted in green, the parts added to the manuscript are highlighted in yellow
Thank you for the opportunity to review your review on endocarditis periodontitis and decay.
Your methods to find relevant scientific reviews are appropriate but needs more details for the exlusion of the last 40 papers.
More details have been added for excluding the last 40 articles.
A full-text assessment was subsequently conducted. During this phase, 44 studies were excluded for the following main reasons:
- 25 did not specifically address the correlation between endocarditis and oral diseases, but rather discussed broader cardiovascular implications or unrelated infections (A);
- 5 lacked sufficient methodological transparency, either due to the absence of protocol registration or the presence of poorly defined protocols and unclear inclusion criteria (B);
- 6 were duplicate systematic reviews or substantial overlaps with previously published reviews, which they effectively updated (C);
- 5 were not systematic reviews, but clinical guidelines, position papers, or survey-based studies (D), table 1.
Table 1. excluded studies, A) studies that did not specifically address the relationship between endocarditis and oral disease, but rather discussed broader cardiovascular implications or unrelated infections;(B) studies that lacked sufficient methodological transparency (no protocol registration or associated with a lack of well-defined protocol or unclear inclusion criteria);(C) studies that were duplicate systematic reviews or substantial overlaps with previously published reviews that represent an update;(D) studies that were not systematic reviews, but clinical guidelines or position papers or questionnaires.
|
Author, data, reference |
Main Topics |
reasons for exclusion |
|
Dave and Tattar, 2025 [14] |
bacterial antimicrobial resistance |
A |
|
Zorman et al., 2025 [15] |
incidence of bioprosthetic mitral valve thrombosis |
A |
|
Barda et al., 2024 [16] |
Intravenous Treatment for Infective Endocarditis |
A |
|
Sonaglioni et al., 2024 [17] |
Echocardiographic Assessment of Mitral Valve Prolapse |
A |
|
Mourad et al., 2025 [18] |
Oral Versus Intravenous Antibiotic Therapy for Staphylococcus aureus Bacteremia or Endocarditis |
A |
|
Veldurthy et al., 2025 [19] |
Incidence of Infective Endocarditis Post Transcatheter Pulmonary Valve Replacement |
A |
|
Monaci et al., 2024 [20] |
Infective Endocarditis Subsequent to Transcatheter Versus Surgical Aortic Valve Replacement |
A |
|
Brown et al., 2024 [21] |
Treatment of endocarditis in people who inject drugs |
A |
|
Nor et al., 2023 [22] |
Systemic Lupus Erythematosus and Cardiovascular Diseases |
A |
|
Diz Dios et al., 2023 [23] |
Prophylaxis guidelines for patients with high-risk cardiac conditions |
D |
|
Lean et al., 20023 [24] |
Prophylactic antibiotic use for infective endocarditis |
A |
|
Budea et al., 2023 [25] |
Clinical characteristics of Infective Endocarditis in older adults and the risk factors that could lead to adverse outcomes |
A |
|
Araújo Júnior et al., 2022[26] |
Presence of Amoxicillin-Resistant Streptococci in the mouth of children |
A |
|
Hussein et al., 2022 [27] |
Quality appraisal of antibiotic prophylaxis guidelines |
D |
|
Shimizu et al.,2022 [28] |
Infection Route of Parvimonas micra |
B |
|
Alifragki et al., 2022 [29] |
Infective Endocarditis by Pasteurella Species |
B |
|
Sindon et al., 2022 [30] |
Health risks for body pierced |
A |
|
Talha et al., 2021 [31] |
Temporal Trends of Infective Endocarditis |
A |
|
Wald-Dickler et al., 2022[32] |
Comparison of intravenous versus oral regimens |
A |
|
Franconieri et al., 2021 [33] |
Rothia spp. infective endocarditis |
B |
|
Herrera-Hidalgo et al., 2020 [34] |
Enterococcus faecalis Endocarditis |
A |
|
Martí-Carvajal et al., 2020 [35] |
A comparison of different antibiotic regimens |
A |
|
Cummins et al., 2020 [36] |
Dentists' knowledge of the relevant guidelines for prescribing antibiotic prophylaxis for the prevention of infective endocarditis |
D |
|
Vahabi et al., 2019 [37] |
review the infective endocarditis literature published or presented from Turkey |
A |
|
Khan et al., 2020 [38] |
Infective endocarditis post-transcatheter aortic valve implantation |
A |
|
Napolitani et al., 2019 [39] |
Kocuria kristinae |
A |
|
Nakatani et al., 2019 [40] |
Guideline on Prevention and Treatment of Infective Endocarditis |
D |
|
Abarbanell et al., 2019 [41] |
Contraceptive use among women with congenital heart disease |
A |
|
Singh Gill et al., 2018 [42] |
Antibiotic Prophylaxis in Dental Implants and Extraction Procedures |
B |
|
Russell et al., 2018 [43] |
Rheumatic Heart Disease |
A |
|
Al-Omari et al., 2014 [44] |
The role of oral antibiotic therapy in treating infective endocarditis |
A |
|
Glenny et al., 2013 [45] |
prophylaxis of bacterial endocarditis in dentistry |
C |
|
Esposito et al. 2013 [46] |
antibiotics at dental implant placement |
A |
|
Tepper et al., 2010 [47] |
Safety of contraceptive |
A |
|
Swedish Council on Health Technology , 2010 [48] |
Antibiotic Prophylaxis for Surgical Procedures |
D |
|
Esposito et al., 2010 [49] |
antibiotics at dental implant placement |
C |
|
Oliver et al., 2008 [50] |
prophylaxis of bacterial endocarditis in dentistry |
C |
|
Esposito et al., 2008 [51] |
antibiotics at dental implant placement |
C |
|
Lockhart et al., 2007 [52] |
efficacy of antibiotic prophylaxis in dental practice |
B |
|
Oliver et al., 2004 [53] |
prophylaxis of bacterial endocarditis in dentistry |
C |
|
Esposito et al., 2003 [54] |
antibiotics at dental implant placement |
C |
How many studies were included by the included reviews? Propably some studies were included in several of them? (BIAS)
The following sections have been added to the text to better detail the number of studies included in the systematic reviews
In the 11 included systematic reviews, a total of approximately 157 primary studies were analyzed. However, due to overlaps between reviews, many of these studies were included in more than one review. For example, common references were identified in the reviews by Sperotto et al. 2024 [67], Cahill et al., 2017 [161] and Lafaurie et al. 2019 [126], González Navarro et al., 2017 [50] and others in particular regarding the incidence of bacteremia and the effects of antibiotic prophylaxis. This overlap introduces a potential source of bias, as some studies may be over-represented in the evidence synthesis. Fortunately, the included reviews provided detailed lists of the primary studies analyzed, limiting the possibility of precise quantification of duplications as the total number of primary studies included in the review was 103 (Lacassin et al., 1995 [58], Strom et al., 1998 [59], Porat et al., 2008 [60], Chen et al., 2015 [61], Chen et al., 2018 [62], Tubiana et al., 2017 [63], Thornhill et al., 2023 [64], Thornhil et al., 2022 [65], Thornhill et al., 2024 [66], Keller et al., 2017 [68],van den Brink et al., 2017 [69], Bates et al., 2017 [70], Sakai Bizmark et al., 2017 [71], Garg et al., 2019 [72], Quan et al., 2020 [73], Vähäsarja et al., 2020 [74], Bikdeli et al., 2013 [75], DeSimone et al., 2015 [76], Toyoda et al., 2017 [77], Sun et al., 2017 [78], Thornhill et al., 2022 [79], Rogers et al., 2008 [80], Pasquali et al., 2015 [81],Pant et al., 2015 [82], Thornhill et al., 2017 [83], DeSimone et al., 2021 [84], Mackie et al., 2016 [85], Knirsch et al., 2020 [86], Weber et al., 2022 [87], Krul et al., 2015 [88], Duval et al., 2012 [89], Dayer et al., 2015 [90], Shah et al., 2020 [91], Zegri-Reiriz et al., 2018 [93], Thornhill et al., 2018 [94], Khairat, 1966 [96], Shanson et al., 1985 [97], Maskell et al., 1986 [98], Shanson et al., 1987 [99], Vergis et al., 2001 [100], Lockhart et al., 2004 [101], Diz Dios et al., 2006 [102], Abu-Ta'a et al., 2008 [103], Anitua et al., 2008 [104], Lockhart et al., 2008 [105], Asi et al., 2010 [106], Esposito et al., 2010 [107], Siddiqi et al., 2010 [108], Chandramohan et al., 2011 [109], Maharaj et al., 2012 [110], DuVall et al., 2013 [111], Limeres Posse et al,. 2016 [112], Van der Meer et al., 1992 [115], Pourmoghaddas et al. 2018 [117], Ali et al., 2017 [118], Cantekin et al., 2015 [119], Cantekin et al., 2013 [120], Suma et al., 2011[121],Siahi-Benlarbi et al., 2010 [122], da Fonseca et al., 2009 [123], Tasioula et al., 2008 [124], Lockhart et al., 2019 [125], Snanson et al., 1978 [127], Roberts et al., 1987 [128], Hall et al., 1993 [129], Hall et al., 1996 [130], Wahlmann et al., 1999 [131] de Souza et al., 2016 [133], Deppe et al., 2007 [134], Wu et al., 2008 [135], Hakeberg et al., 1999 [136], Nakamura et al., 2011 [137], Bratel et al., 2011 [138], Lockhart eta al., 2009 [140], Bahrani-Mougeot et al., 2008 [141], Heimdahl et al., 1980 [142], Rajasuo et al., 2004 [143], Roberts et al., 2006 [144], Tomás et al., 2007 [145], Roberts et al., 1998 [146], Benítez-Páez et al., 2013 [147], Maharaj et al., 2012 [148], Peterson et al., 1976 [149], Rahn et al., 1995 [150], Sefton et al., 1990 [151], Hall et al., 1996 [152], Tuna et al., 2012 [153], Sweet et al., 1978 [154], Cannell et al., 1991 [155],Aitken et al., 1995 [156], Wahlmann et al., 1999 [157], Josefsson et al., 1985 [158], Tomás et al., 2008 [159], Piñeiro et al., 2010 [160], Imperiale and Horwitz, 1990 [162], Baltch et al., 1982 [163], Coulter, et al., 1990 [164], Head et al., 1984 [165], Shanson et al., 1978 [166], Horstkotte et al., 1987 [167], Thornhill et al., 2011 [168], Desimone et al., 2012 [169] and Salam et al., 2014 [170])
The outcomes you looked at should be clearly appointed. The results of the outcomes could be presented in a separate figure or table. Preventive antibiotic usage is an outcome in only 4/5 included papers. Please dicuss the different antibiotic regimes more in detail. What evidence did you find on the use of antibiotics for caries and periodontal treatment and prevention of endocarditis? Did you find any differences in studies included where severe periodontitis cases (stage 3+4) were treated? Did you find any age dependency in the includes studies?
We thank the reviewer for this insightful and constructive comment. In response, we have made the following revisions to improve the clarity and completeness of our manuscript: all the issues raised have been expanded and added to the manuscript in the results section. a table 3 with the major results and conclusions obtained from the included reviews has also been added. the portions added to the manuscript are reported below
The primary outcomes evaluated in the included systematic reviews were:
- Incidence of infective endocarditis (IE) following dental procedures
- Association between oral conditions (periodontitis and dental caries) and IE
- Bacteraemia induced by dental procedures
- Effectiveness of antibiotic prophylaxis (AP) in preventing IE
- Antimicrobial resistance patterns
- Impact of oral health interventions in high-risk cardiac patients
These outcomes are summarised in Table 3 and further discussed in the discussion section. In total, 8 out of 11 reviews included outcomes related to antibiotic prophylaxis, while 4 out of 11 specifically quantified its effectiveness. Only 1 out of 11 reviews included data on caries-related infections, focusing specifically on children with congenital heart disease (CHD).
Among the reviews that reported data on antibiotic prophylaxis (AP), the most commonly studied regimens included oral or intravenous amoxicillin with clavulanic acid, and oral azithromycin in penicillin-allergic patients. According to Lafaurie et al. (2019), azithromycin showed greater efficacy in reducing bacteraemia compared to clindamycin. Amoxicillin was typically administered as a single 2 g oral dose 30–60 minutes prior to the procedure, whereas intravenous regimens were reserved for hospitalised or high-risk patients undergoing surgery.
Barbosa-Ribeiro et al. (2024) observed high levels of resistance to clindamycin, metronidazole, and rifampicin in E. faecalis, highlighting the importance of local antibiogram data. Only 4 of the included reviews provided direct comparisons between antibiotic regimens, and no clear consensus emerged from the literature regarding the optimal agent or dosage.
Preventive antibiotics were discussed primarily in relation to invasive procedures, and not as a routine treatment for caries or mild periodontal therapy.
Regarding caries, only one review (Karikoski et al., 2021) addressed this condition, showing that children with congenital heart disease (CHD) have a significantly higher prevalence of dental caries. However, this review did not evaluate the role of antibiotics in caries management as a preventive strategy for IE. No systematic review supported the routine use of antibiotic therapy for non-invasive caries treatment (e.g., restorations).
As for periodontal treatment, several reviews (Kussainova et al., 2025; Lafaurie et al., 2019; González Navarro et al., 2017) considered procedures such as scaling and periodontal surgery in the context of bacteraemia risk. However, findings consistently showed no significant association between non-invasive periodontal treatment and an increased risk of IE (e.g., Kussainova et al. reported an OR of 0.69; p = 0.41 for periodontal therapy). Consequently, routine antibiotic prophylaxis for periodontal therapy is not supported by current evidence, unless the procedure is invasive and the patient is considered high-risk.
Age-related findings were limited. Only one review (Karikoski et al., 2021) specifically focused on children with congenital heart disease and reported a higher caries burden and associated IE risk. Other reviews included populations with heterogeneous age ranges, but did not stratify findings by age group, making it difficult to assess whether age modifies the association between oral diseases and IE.
Table 2. Summary of Results from the Included Systematic Reviews on Infective Endocarditis, Oral Conditions, and Antibiotic Prophylaxis.
|
Autor data reference |
|
|
Kussainova et al., 2025 [57] |
Kussainova et al., 2025 found an association between IE and invasive dental procedures (OR 1.49, 95% CI 1.25–1.76; p < 0.00001) while subgroup analysis showed an increased risk of IE following tooth extraction (OR 2.73, 95% CI 1.46–5.11; p = 0.002) and oral surgery (OR 6.33, 95% CI 2.43–16.49; p = 0.0002) in high-risk patients. They also identified the strongest association between IE and tooth extraction (OR 1.90, 95% CI 1.17-3.08; p = 0.010; I2 = 80%;) and oral surgery (OR 3.11, 95% CI 1.20-8.05; p = 0.02; I2 = 77%;) while there was no significant association between IE and invasive dental procedures such as scaling (OR 1.00, 95% CI 0.85-1.18; p = 1.00; I2 = 0%;), endodontic treatment (OR 1.04, 95% CI 0.73-1.49; p = 0.82; I2 = 0%;) and periodontal treatment (OR 0.69, 95% CI 0.28-1.67; p = 0.41; I2 = 69%; ).
|
|
Sperotto et al. 2024 [67] |
Sperotto et al. 2024 found that antibiotic prophylaxis was associated with a significantly lower risk of infective endocarditis after invasive dental procedures in high-risk individuals (pooled RR, 0.41; 95% CI, 0.29-0.57; P for heterogeneity = 0.51; I2, 0%) |
|
Friedlander and Couto-Souza, 2023 [92] |
Friedlander and Couto-Souza, 2023 To investigate whether the European Society of Cardiology (ESC) guidelines, which recommend AP only for “high-risk” patients, are also appropriate for patients with valvular heart disease defined as “intermediate risk” (such as bicuspid aortic valve – BAV – or mitral valve prolapse – MVP). They found that patients with BAV or MVP, traditionally classified as “intermediate risk” according to current ESC guidelines, may actually be at significantly higher risk of infective endocarditis (IE) and related complications. Based on large-scale data from the Netherlands, Spain and the UK, the authors reported a higher incidence of IE, hospital mortality and intracardiac complications in these patients, supporting a reclassification of BAV and MVP as “high-risk” conditions requiring prophylactic antibiotic coverage before high-risk dental procedures. |
|
Albakri et al., 2022[95] |
Albakri et al. (2022) compared AP with placebo. The results showed that AP significantly reduced the incidence of bacteremia by 49% (risk ratio 0.51; 95% CI: 0.45-0.58; P = 0.0001). Although bacteremia has been used as a surrogate marker for IE, the authors acknowledged that direct evidence linking AP to a reduction in IE remains inconclusive due to the rarity of the disease and the impracticality of conducting large-scale studies. |
|
Bergadà-Pijuan et al., 2023 [113] |
Bergadà-Pijuan et al. (2022) conducted a systematic review to assess the efficacy of AP in IE in adults undergoing dental procedures. Of 264 publications reviewed, only one prospective cohort study (Tubiana et al., 2017) met the inclusion criteria. This study focused exclusively on high-risk patients with prosthetic heart valves. The results showed a non-significant reduction in the incidence of IE when AP was administered (RR 0.39; p = 0.11), suggesting a potential protective effect, although inconclusive. The authors concluded that the evidence supporting or disapproving AP remains weak, particularly for low- and moderate-risk populations, and highlighted the urgent need for well-powered clinical trials to clarify current guideline recommendations. |
|
Rutherford et al., al., 2022 [114] |
The systematic review by Rutherford et al. (2022) aimed to assess whether AP before invasive dental procedures reduces the incidence of bacterial endocarditis in at-risk individuals. Despite extensive database searches, only one eligible case-control study was included (Van der Meer et al., 1992). This study found no statistically significant difference in the incidence of bacterial endocarditis between patients receiving penicillin prophylaxis and those not receiving penicillin prophylaxis (OR 1.62; 95% CI: 0.57–4.57). The overall certainty of the evidence was judged to be very low and no data were available regarding mortality, adverse events or cost-effectiveness. The authors concluded that there is still no reliable evidence to support or refute the effectiveness of AP in preventing BE after dental procedures in high-risk populations and that clinical decisions should be guided by shared decision-making and discussion of potential benefits and risks. |
|
Karikoski et al., 2021 [116] |
Karikoski et al. (2021) conducted a systematic review comparing the prevalence of dental caries in children with congenital heart disease (CHD) compared to healthy peers. Of the nine included studies, seven reported a higher prevalence of caries in children with CHD, with statistically significant differences in three. Mean dmft and DMFT scores were consistently higher in the CHD group. For example, in one study, 77.4% of children with CHD had caries in primary teeth (dmft > 0) compared to 56.5% of controls. Evidence suggests that children with complex or surgically treated CHD (severity grades 2–4) are at particular risk. Despite heterogeneity and limitations in study quality and design, the review concluded that children with CHD have a higher caries burden, highlighting the need for targeted prevention strategies in this vulnerable population. |
|
Lafaurie et al., 2019 [126] |
Lafaurie et al. (2019) conducted a review to evaluate the effectiveness of AP in reducing bacteraemia after tooth extractions. Pooled results showed that AP reduced the incidence of bacteraemia by 50% (RR 0.50; 95% CI: 0.38-0.67). Amoxicillin, azithromycin and clindamycin (AHA protocol) demonstrated variable efficacy, with the intravenous amoxicillin-clavulanic acid combination achieving the greatest reduction (RR 0.01). |
|
Lockhart rt al., 2019 [132] |
Lockhart et al. (2019) conducted a review to assess whether professional dental treatment before heart valve surgery (CVS) or LVAD implantation reduces postoperative complications. Results showed no statistically significant differences in key outcomes, including all-cause mortality (RR 1.00; 95% CI: 0.53-1.91), infective endocarditis (RR 1.30; 95% CI: 0.51-3.35), postoperative infections (RR 1.01; 95% CI: 0.76-1.33), or length of hospital stay (mean difference +2.9 days; 95% CI: -2.3 to 8.1). The certainty of the evidence was assessed as very low due to risk of bias and imprecision. The authors concluded that current data do not support a definitive benefit or harm of preoperative dental treatment in adults undergoing CVS, emphasizing the need for individualized, interdisciplinary decision making. |
|
González Navarro et al., 2017 [139] |
González Navarro et al. (2017) evaluated the duration and extent of bacteremia after oral surgery and the potential impact of AP. The review confirmed that, although AP does not completely prevent post-procedure bacteremia, it significantly reduces its extent and persistence. Amoxicillin was the most frequently studied antibiotic, followed by clindamycin, erythromycin, teicoplanin, and others. The most commonly isolated microorganism was Viridans streptococcus. Despite the variability of the results, high-dose systemic amoxicillin (e.g., 2–3 g 1 h before surgery) was associated with the most consistent protective effect. However, several studies reported no significant benefit compared to placebo. The authors concluded that, although AP may reduce the risk of IE in high-risk subjects, standardized protocols and further high-quality studies are needed to clarify the clinical utility of AP in dental surgery. |
|
Cahill et al., 2017 [161] |
Cahill et al. (2017) conducted a review to assess the effectiveness of AP in preventing IE following dental procedures. While AP significantly reduced the incidence of post-procedure bacteraemia (RR 0.53; 95% CI: 0.49-0.57), the evidence supporting a protective effect against IE was inconclusive. Observational studies showed a non-significant trend in favour of AP (OR 0.59; 95% CI: 0.27-1.30; p = 0.14) and only one time trend study (from the UK) reported an increase in the incidence of IE following complete cessation of AP. The authors concluded that the current evidence base is limited by methodological heterogeneity and a lack of randomized clinical trials, but recommended that PA may remain a justified low-risk intervention for high-risk individuals, in accordance with existing ESC and ACC/AHA guidelines. |

Reviewer 4 Report
Comments and Suggestions for Authors
Dear authors,
congratulations on your effort! You've done a great job!
Author Response
We sincerely thank the Reviewer for the positive feedback and encouraging words. We truly appreciate the time dedicated to reviewing our work
Round 2
Reviewer 2 Report
Comments and Suggestions for Authors
I would like to thank the authors for the thoughtful and appropriate revisions made to the manuscript. Although the study investigates a field of oral medicine that is already well established, the analysis of the existing literature is nonetheless insightful and represents a valuable contribution to ongoing research.
However, given that the structure of the manuscript more closely resembles that of a systematic review, I would recommend revising the classification of the paper accordingly, and including an assessment of the risk of bias among the included studies. Alternatively, should the authors prefer to maintain the current classification, I suggest adapting the manuscript to more closely align with the standard structure of a narrative review.
Author Response
Answer
We sincerely thank the Reviewer for their kind and constructive feedback. In response to the valuable recommendation, we have revised the manuscript to better reflect the characteristics of a systematic review, as initially suggested. Accordingly, we have now included a structured assessment of the risk of bias for the included systematic reviews using the ROBIS tool, which is specifically designed for evaluating the quality of systematic reviews.
This assessment is detailed in the new Section 2.6 (Risk of Bias) within the Methods, and the findings are presented in Section 3.1 (Risk of Bias) of the Results, as well as summarized in Table 4. The evaluation was conducted independently by two authors (M.D. and A.B.), following the ROBIS guidelines, and focused on key domains such as study eligibility criteria, identification and selection of studies, and data synthesis.
We believe that this addition enhances the methodological robustness and transparency of the study, aligning the structure more clearly with the standards of a systematic review, as per the Reviewer’s helpful suggestion.The following parts have been added to the manuscript
2.6. Risk of Bias
The risk of bias in the individual systematic reviews was assessed by two authors (M.D. and A.B.). The ROBIS (Risk of Bias in Systematic Reviews) was used as an assessment tool specifically developed to assess the risk of bias in systematic reviews. Studies with a high risk of bias were excluded from the review [14].
The distinction between bias within the review process (meta-bias) and bias in the primary studies included in the review is critical. A systematic review may be considered at low risk of bias even if all the included primary studies are at high risk of bias, provided that the review appropriately assesses the risk of bias in the primary studies before drawing its conclusions [14].
……………………………………………………………………………………………………..…………………………………………………………………………………………………..
3.1. Risk of Bias
The risk of bias for systematic reviews was determined using the ROBIS tool, and for each factor, it was evaluated as “low”, “high”, or “unclear”. The three phases of the evaluation process were as follows: Phase 1: the evaluation of the relevance of the research question (PICO); Phase 2: the identification of critical points of the review process; and Phase 3: the evaluation of the overall risk of bias of the review. All data related to the risk of bias are reported in Table 4.
Table 4. Risk of bias, ROBIS scale: ok (low);? (unclear).
|
First Author, Data |
Phase 1 |
Phase 2 |
Phase 3 |
|||
|
PICO |
Study Eligibility Criteria |
Identification and Selection of Studies |
Data Collection and Study Appraisal |
Synthesis and Findings |
Risk of Bias in the Review |
|
|
Kussainova et al., 2025 [58] |
ok |
ok |
ok |
ok |
ok |
ok |
|
Sperotto et al. 2024 [68] |
ok |
ok |
? |
ok |
ok |
ok |
|
Friedlander and Couto-Souza, 2023 [93] |
|
? |
? |
? |
OK |
Ok |
|
Albakri et al., 2022[96] |
ok |
ok |
ok |
? |
ok |
ok |
|
Bergadà-Pijuan et al., 2023 [114] |
ok |
ok |
ok |
ok |
ok |
ok |
|
Rutherford et al., al., 2022 [115] |
ok |
ok |
ok |
ok |
ok |
ok |
|
Karikoski et al., 2021 [117] |
ok |
ok |
? |
ok |
ok |
ok |
|
Lafaurie et al., 2019 [127] |
ok |
ok |
ok |
ok |
ok |
ok |
|
Lockhart rt al., 2019 [133] |
ok |
ok |
ok |
ok |
ok |
ok |
|
González Navarro et al., 2017 [140] |
ok |
ok |
? |
? |
ok |
ok |
|
Cahill et al., 2017[162] |
ok |
ok |
? |
? |
ok |
ok |
The main critical issues related to the individual revisions are as follows:
- Sperotto et al. 2024 [68]: Identification and Selection of Studies (?):The start or end dates of the review were not specified.
- Friedlander and Couto-Souza, 2023 [93]: Study eligibility criteria (?): The protocol number with which the systematic review was registered was not reported; Data collection and study appraisal (?): The risk of bias was not formally assessed using an appropriate scale or tool; Identification and selection of studies (?): The selection was performed only on one database (PubMed).
- Albakri et al., 2022[96]: Data collection and study appraisal (?): The risk of bias was not formally assessed using an appropriate scale or tool;
- Karikoski et al., 2021 [117]: Identification and Selection of Studies (?):The systematic review by Karikoski et al. (2021) did not report registration of a review protocol in any public registry.
- González Navarro et al., 2017 [140]: Identification and Selection of Studies (?):The systematic review by González Navarro et al., 2017 did not report registration of a review protocol in any public registry; Data collection and study appraisal (?): The risk of bias was not formally assessed using an appropriate scale or tool;
- Cahill et al., 2017 [162]: Identification and Selection of Studies (?):The systematic review by Cahill et al., 2017 did not report registration of a review protocol in any public registry; Data collection and study appraisal (?): The risk of bias was not formally assessed using an appropriate scale or tool;

Reviewer 3 Report
Comments and Suggestions for Authors
Thank you for the resubmission. Most of the methods-topics are now clearly presented.
But again the suggestion not to use an endocarditis prophylaxis for treatment with non-surgical periodontal treatment in patients at risk for endocardititis, need to be explained and discussed in more detail. Otherwise its very missleading to international endocarditis prevention regimes in cardiological patients mit endocarditis risk. In many countries endocarditis prophylaxis is recommended with amoxycillin without clavulanis acid. This topic also needs discussion in this review.
Author Response
ANSWER
We thank the Reviewer for this important remark. In response, we have thoroughly revised the manuscript to include a dedicated section discussing the indication of endocarditis prophylaxis in high-risk patients undergoing non-surgical periodontal treatment. We clarified that these procedures may induce transient bacteremia and, according to AHA and ESC guidelines, may warrant prophylaxis in specific cases. Furthermore, we addressed the use of amoxicillin without clavulanic acid as the recommended first-line agent, based on international protocols and current evidence.
below are the parts added to the manuscript
4.5. Endocarditis Prophylaxis in At-Risk Patients Undergoing Non-Surgical Periodontal Treatment: Role of Amoxicillin
The use of AP to prevent IE in patients undergoing dental procedures remains a subject of ongoing debate, particularly in the context of non-surgical periodontal treatments such as scaling and root planing. Although a direct causal relationship between procedure-induced transient bacteremia and the onset of IE has not been definitively established, a substantial body of evidence supports the notion that manipulations involving gingival or periapical tissues may introduce oral bacteria into the bloodstream, posing a potential risk for individuals with specific predisposing cardiac conditions (Albakri et al., 2022[96] Rutherford et al., al., 2022 [115] Lafaurie et al., 2019 [127])
4.5.1. Risk Associated with Non-Surgical Periodontal Procedures
Several studies have demonstrated that non-surgical periodontal procedures—particularly in patients with active gingival inflammation or periodontitis—can induce bacteremia at frequencies comparable to those observed following dental extractions. For instance, the study by Albakri et al., 2022 [96] reports that periodontal probing and scaling were associated with bacteremia rates ranging from 10% in patients with gingivitis to 40% in those with periodontitis, while tartar removal (i.e., supragingival and subgingival scaling) resulted in a bacteremia prevalence of 24.5%. According to Albakri et al. [96], these variations are influenced by the severity of the underlying periodontal disease.
Conversely, the meta-analysis by Kussainova et al., 2025 [58] found no significant association between invasive dental procedures and the development of infective endocarditis. Specifically, the pooled odds ratios (OR) were as follows:Scaling: OR 1.00, 95% CI 0.85–1.18; p = 1.00; I² = 0%;Endodontic treatment: OR 1.04, 95% CI 0.73–1.49; p = 0.82; I² = 0%; Periodontal treatment: OR 0.69, 95% CI 0.28–1.67; p = 0.41; I² = 69%.
Among the studies included in the Kussainova review [58], only Lacassin et al., 1995 [59] reported a trend toward increased risk associated with scaling and root canal therapy.
These findings indicate that even non-surgical procedures such as scaling and periodontal probing can induce transient bacteremia, which represents a theoretical risk for infective endocarditis in predisposed patients. Consequently, such interventions may reasonably be classified as invasive procedures in the context of IE prevention for high-risk individuals.
4.5.2. International Guidelines and Indications for Prophylaxis (Amoxicillin)
Current guidelines from the American Heart Association (AHA)[207] and the European Society of Cardiology (ESC)[208] recommend AP exclusively for patients at high risk—such as individuals with prosthetic heart valves, a history of IE, or complex congenital heart disease—who undergo dental procedures involving manipulation of gingival tissues or the periapical region of teeth. Non-surgical periodontal treatments, particularly in the presence of active inflammation or gingival bleeding, clearly fall within this definition.
Most international protocols recommend a single dose of 2 g of oral amoxicillin without clavulanic acid, administered 30–60 minutes prior to the procedure, as reported in trials included in the reviews by Rutherford et al., al., 2022 [115] , Lafaurie et al., 2019 [127] and Bergadà-Pijuan et al., 2023 [114].
While it is well established that antibiotic prophylaxis reduces post-procedural bacteremia—as shown in the systematic review by Albakri et al., 2022 [96], which reported an approximate 49% reduction in bacteremia risk (Risk Ratio 0.51, 95% CI 0.45–0.58; p < 0.0001)—other studies suggest that this reduction does not necessarily translate into a decreased risk of developing IE. Supporting this view is the Cochrane systematic review by Rutterford et al. (2022) [115], which included non-surgical periodontal procedures (such as supragingival and subgingival scaling, as well as curettage) among the evaluated interventions, all of which are potentially associated with transient bacteremia.
In detail, Rutterford et al. evaluated the administration of amoxicillin (2–3 g) or other beta-lactam antibiotics, 30–60 minutes before the procedure, compared with placebo or no prophylaxis in patients at high risk for IE. The only study included in the review was Van der Meer et al., 1992 [116], which found that prophylaxis did not result in a statistically significant reduction in the risk of endocarditis among patients undergoing scaling or similar treatments (OR 1.62; 95% CI 0.57–4.57). However, the detailed analysis of associated procedures revealed that 13% of IE cases had undergone subgingival scaling, 25% had received scaling combined with polishing (root surface debridement), and notably, 43% of control patients underwent the same procedures without developing IE. These findings, while not conclusive, highlight a possible association between such interventions and bacteremia, even in the absence of demonstrated efficacy of antibiotic prophylaxis in preventing endocarditis in this context.
It is also important to note that oral amoxicillin combined with clavulanic acid has not shown any additional benefit in this setting [116].
In patients with penicillin allergy, oral azithromycin appears to be more effective than clindamycin in reducing bacteremia, as reported by Lafaurie et al., 2019 [127]. Specifically, Lafaurie et al. documented that pre-procedural use of amoxicillin, azithromycin, and clindamycin (as per AHA protocols) led to reductions in bacteremia risk of 59%, 49%, and 11%, respectively, compared to no prophylaxis. Aggregated estimates also showed that two antibiotics not included in AHA protocols—namely moxifloxacin and intravenous amoxicillin–clavulanic acid—were associated with significant reductions in bacteremia following invasive dental procedures, by 41% and 99%, respectively [127]

Round 3
Reviewer 3 Report
Comments and Suggestions for Authors
Thank you for your answers on my questions and the new sections in your paper.